# Faster Optimal Univariate Microgaggregation

**Felix I. Stamm**                                                    *felix.stamm@cssh.rwth-aachen.de*
*RWTH Aachen University*
*Germany*

**Michael T. Schaub**                                                    *schaub@cs.rwth-aachen.de*
*RWTH Aachen University*
*Germany*

**Reviewed on OpenReview:** *https: // openreview. net/ forum? id= s5lEUtyVly*

## Abstract

Microaggregation is a method to coarsen a data set, by optimally clustering data points in groups of at least $k$ points, thereby providing a $k$-anonymity type disclosure guarantee for each point in the data set. Previous algorithms for univariate microaggregation had a $O(kn)$ time complexity. By rephrasing microaggregation as an instance of the concave least weight subsequence problem, in this work we provide improved algorithms that provide an optimal univariate microaggregation on sorted data in $O(n)$ time and space. We further show that our algorithms work not only for sum of squares cost functions, as typically considered, but seamlessly extend to many other cost functions used for univariate microaggregation tasks. In experiments we show that the presented algorithms lead to performance improvements on real hardware.

## 1  Introduction

Public data, released from companies or public entities, are an important resource for data science research. Such data can be used to provide novel types of services to society and provide an essential mechanism for holding public or private entities accountable. Yet, the benefits of publicly released data have to be carefully traded off against other fundamental interest such as privacy concerns of affected people (Aggarwal & Yu, 2008). A common practical solution to circumvent these problems is thus to reduce the resolution of the collected raw data, e.g., via spatial or temporal aggregation, which leads to a coarse-grained summary of the initial data.

Microaggregation (Defays & Anwar, 1998; Samarati, 2001; Domingo-Ferrer & Mateo-Sanz, 2002; Domingo-Ferrer & Torra, 2005) is a method that aims to provide an adaptive optimal aggregation that remains informative, while providing a certain guaranteed level of privacy. Specifically, in microaggregation we aim to always group at least $k$ data items together onto a common representation. For instance, for vectorial data, we may map data points to their centroid values, while ensuring a cluster size of at least $k$ points. The found centroid values of the clusters may thus be released and used as aggregated representation of the original data. Clearly, for the aggregated data to remain useful for further analysis, it should remain closely aligned with the original data according to some metric relevant for the task at hand. Hence, in microaggregation we seek to minimize a distortion metric of the aggregated data, while respecting the minimal group size constraint. This minimal group-size constraint (at least $k$ data-items per group), which ensures a guaranteed minimal level of "privacy" within each cluster, is a major difference to standard clustering procedures such as $k$-means, which simply aim to obtain a low-dimensional representation of the observed data in terms of clusters.

Microaggregation tasks are usually formulated as discrete optimization problems, where low costs indicate high similarity among the data items in each cluster. Similar to many other clustering problems, microaggregation is in general an NP-hard problem (see Oganian & Domingo-Ferrer (2001) or appendix B.1). A specific form

Figure 1: Example application of univariate microaggregation to a toy medical data set. Left we have the original data containing quasi identifying information such as age and sensitive information such as health status. In the anonymized data an attacker interested in the health status of an individual and equipped with knowledge about the age of an individual can no longer infer the health status with certainty. To achieve such an anonymization, potentially identifying attributes such as age are aggregated. The task of finding such an aggregation while minimizing the distortion of the data is called univariate microaggregation. In this work, we are concerned with efficient algorithms to solve univariate microaggregation for various kinds of distortion metrics/cost functions.

of microaggregation that remains efficiently solvable, however, is univariate microaggregation (UM), in which the data points $\{x_i\}$ to be aggregated are (real-valued) scalars. This is the focus of our work here.

Despite its relative simplicity, univariate micro-aggregation is a useful primitive in a number of problems. A concrete example is the anonymization of degree sequences of graphs (Casas-Roma et al., 2017). Further, univariate microaggregation can also be used to provide heuristic solutions to microaggregation tasks in higher dimensions with the help of projections (Domingo-Ferrer & Mateo-Sanz, 2002). In this work we consider the univariate microaggregation problem under a wide range of cost functions, including the typically considered sum-of-squares error (SSE) based on the (squared) $\ell_2$ norm, the $\ell_1$ norm, the $\ell_\infty$ norm and round-up cost error types.

**Contributions** Our main contributions are as follows (see Figure 2 for a visual representation).

- We show that for "ordered minimizable" cost functions UM can be solved in $O(n^2)$, improving previous results with exponential runtime. We achieve this by rephrasing UM as a least weight subsequence problem (Wilber, 1988; Wu, 1991).

- We show that for ordered minimizable cost functions, which in addition have a "splitting is beneficial" property, this can be improved to $O(kn)$. This includes many popular cost functios based on the $\ell_1$ norm (sum of absolute errors), $\ell_2$ norm (sum of squared errors), $\ell_\infty$ (maximal error) norm, and round up/down cost functions.

- Finally, we show that for ordered minimizable cost functions which are concave, algorithms with $O(n)$ time and space are possible. These findings apply to above mentioned examples of $\ell_p$ norm-based cost functions.

In our presentation we focus on three key conceptual ideas: "ordered minimizable", "splitting is beneficial", and concave costs. While these concepts are already implicitly present in the literature, they are typically not made explicit, which can make it difficult to relate the underlying algorithmic ideas to each other. Making these concepts explicit, enables us to unfold a natural problem hierarchy within the associated univariate microaggregation formulations, which leads us to the design of two algorithms that use all three concepts: the simple+ algorithm and the staggered algorithm. These algorithms have a faster runtime compared with previous algorithms for the UM task. Lastly, we provide several practical improvements specific to UM that allow for (i) algorithms that run empirically faster in practice and (ii) substantially decreased impact of floating point errors on the resulting clustering. Overall, our work thus enables to robustly compute

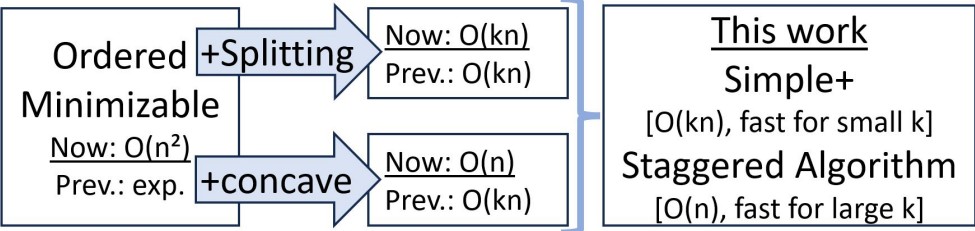

Figure 2: Complexity of UM on sorted data for different classes of cost functions. For ordered minimizable cost functions univariate microaggregation can be solved in $O(n^2)$. If additionally splitting is beneficial, then one can actually solve UM in $O(kn)$ while for concave cost functions UM can be solved in $O(n)$. For cost functions that have both properties we provide the Simple+ and the Staggered algorithm which have faster empirical runtime without sacrificing worst case bounds.

optimal UM-clusterings for various cost functions and large values of $n$ and $k$. An implementation of the presented ideas is available at `https://github.com/Feelx234/microagg1d` and the code can also be found at `https://doi.org/10.5281/zenodo.10459327`.

**Outline** In Section 2 we first outline definitions associated with UM and the related least weight subsequence (LWSS) problem. In Section 3 we then show that for ordered minimizable cost functions, we can reformulate UM as LWSS problem which allows for $O(n^2)$ algorithms. For cost functions with the splitting is beneficial property, this runtime complexity can be improved to $O(kn)$. For concave cost functions this can be futher improved to $O(n)$. Closing Section 3 we introduce a simpler and empirically faster $O(n)$ algorithm for UM. In Section 4 we present considerations for running UM on real hardware. First, we present two strategies to avoid issues arising from finite float precision. Second, we illustrate a small algorithmic trick to use fewer instructions when computing the cluster cost, which is the main computation carried out within UM. We verify that the presented theoretical considerations lead to significant empirical performance improvements in Section 5.

## 1.1 Related Work

The question of how to publicly release data, while retaining certain levels of privacy, has become increasingly important in recent years. Privacy preservation for data sets is most relevant if data entries contain both public as well as sensitive private information. Note that what information is considered public and private can depend on the application scenario. For instance, we may have tabular data, where each row contains public information such as name or age, in conjunction with private information such as medical diagnoses. In this case, an important objective is to design a surrogate data set, such that it is virtually impossible to use public information to deduce private information about an individual.

The public attributes of a database can usually be split into identifiers and quasi-identifiers. To privacy-harden a database it is typically insufficient to merely remove identifiers such as name or social security number. It can often remain possible to identify an individual in the data set by a combination of quasi-identifiers (Rocher et al., 2019), e.g., there is only one 30 year old female in the data. Thus other privacy protection measures need to be employed.

To combat the leakage of sensitive information, different privacy concepts such as $l$-diversity (Machanava-jjhala et al., 2007), $t$-closeness (Li et al., 2007), and the popular $k$-anonymity (Samarati & Sweeney, 1998; Samarati, 2001; Sweeney, 2002) have been proposed. The latter is the privacy concept we are concerned with in this work. If a data set is $k$-anonymous, it is not possible to use any combination of quasi-identifiers to narrow down the set of individuals described by those quasi-identifiers to less than $k$ individuals. Because there will always be $k - 1$ other rows indistinguishable by quasi-identifiers alone, it remains impossible to deduce which data item corresponds to a specific identity, no matter how much knowledge about the quasi-identifiers of an individual is available. Unfortunately, data sets are usually not inherently $k$-anonymous but the released data needs to be altered to become $k$-anonymous distorting the statistics of the released data.

Microaggregation (Domingo-Ferrer & Mateo-Sanz, 2002; Defays & Anwar, 1998; Domingo-Ferrer & Torra, 2005) was proposed as a method to make a data set $k$-anonymous. Therefore data entries are grouped into groups of size at least $k$, and the quasi-identifiers of entries are replaced with their group-centroid values. To uphold the utility of the released data, the grouping is conducted by minimizing a distortion metric subject to a minimal group size constraint. Solving microaggregation with more than one variable, so called multivariate microaggregation, is known to be NP-hard for the sum-of-squares cost function (Oganian & Domingo-Ferrer, 2001), a popular distortion metric. To perform multivariate microaggregation in practice, several algorithms have been proposed. Some of these alogorithms come with approximation guarantees, while others are simply heuristics. We refer to Yan et al. (2022) for many references to algorithms for multivariate microaggregation.

Microaggregation with only one variable, so called univariate microaggregation, is known to be solvable in polynomial time. More precisely, $O(kn)$ for the sum-of-squares cost function (Hansen & Mukherjee, 2003) by constructing a graph and solving a shortest path problem on this graph. Algorithms which solve univariate microaggregation may also be used to solve multivariate microaggregation problems. Recently, other cost functions besides SSE have been used for univariate microaggregation tasks such as degree-sequence anonymisation (Casas-Roma et al., 2017).

We remark that the term univariate microaggregation is also used in the literature to describe a microaggregation problem of *ordered* high dimensional data: In that scenario we are provided with data vectors $x_i \in \mathbb{R}^d$, with $d \geq 2$ and an extrinsically defined total order of the points, which must be respected when creating clusters. Hence, any cluster that contains the points $x_i$ and $x_j$ must also contain all points greater than $x_i$ and smaller than $x_j$ according to the provided ordering. While some of the ideas we develop here may be extended to this scenario, this is in general not a trivial task and beyond the scope of this work.

## 2 Problem Formulation and Preliminaries

In univariate microaggregation (UM) we consider a set of scalar data points $x_i \in \mathbb{R}$ for $i = 1, \ldots, n$. To keep the treatment general we allow for repeated elements $x_i$, i.e., we consider our universe $\Omega = \{\!\{x_1, \ldots, x_n\}\!\}$ to be a multi-set of points. The goal of UM can now be formulated as follows.

**Definition 1** (Univariate Microaggregation problem)**.** *Given a multi-set of scalars $\Omega = \{\!\{x_1, \ldots, x_n\}\!\}$, find a non-empty partition of the points into clusters $X_j$, such that each cluster has size at least $k$ and a total additive cost function $TC(\{\!\{X_1, \ldots\}\!\}) := \sum_j C(X_j)$ is minimized, where $C$ denotes a cost function that measures the distortion within each cluster.*

Formally, we look for a partition, i.e., a split of the multi-set $\Omega$ into nonempty multi-sets $X_i \neq \emptyset$, such that $\bigcup_{i=1}^n X_i = \Omega$. To avoid confusion, in this work the union of two multi-sets is an additive union, i.e., the multiplicities of the elements add up when forming the union of two multi-sets. Notice, that there might be multiple partitions minimizing the total cost even if all $x_i$ are unique. This is usually not a problem as we are typically interested in finding any partition minimizing the total cost.

A brute-force approach, neglecting possible structure in the cost function $C$ needs at least exponential time, as all possible partitions have to be assessed. However, two important observations have been employed in the literature to lower this complexity. First, for many cost functions it can be shown that a minimal cost partition has no interleaved clusters (see Corollary 7 or Domingo-Ferrer & Mateo-Sanz (2002) for the special case of SSE). This substantially restricts the search space that needs to be explored in order to find an optimal solution, as it implies that only pairwise-ordered clusterings need to be considered. From now on we thus assume that data points $x_i$ are sorted in ascending order, i.e. $i < j \Rightarrow x_i \leq x_j$. For a given cost function $C$, this allows us to denote by $C(i, j) := C(\{\!\{x_{i+1}, \ldots, x_j\}\!\})$, the cost of a cluster with starting point $x_{i+1}$ and endpoint $x_j$. Second, for many cost functions, there always exist an optimal cluster assignment in which no cluster has size larger than $2k - 1$, i.e., it is never detrimental to split clusters of size at least $2k$ into smaller clusters (see eq. 2 or Domingo-Ferrer & Mateo-Sanz (2002) for the special case of SSE). This also reduces the size of the search space, as there is no need to check possible clusterings containing clusters of size greater $2k - 1$.

Previously, these observations were used to solve univariate micoraggregation for the SSE cost function on sorted data in $O(kn)$ time (Hansen & Mukherjee, 2003; Mortazavi, 2020). In the following we will exploit the fact that many popular cost functions are concave cost functions, which can be used to create even faster algorithms. For these concave cost functions it is possible to achieve $O(n)$ runtime algorithms for the UM problem on sorted data by rephrasing UM as a least weight subsequence problem.

## 2.1 The Least-Weight Subsequence Problem

As we will see in the next sections, univariate microaggregation can actually be phrased such that established algorithms which solve the least-weight subsequence problem can be applied to UM. Hence, we briefly recall the relevant aspects of the least-weight subsequence problem

**Definition 2** (Least weight subsequence problem (Wilber, 1988) )**.** *Given an integer $n$ and a cost function $C(i, j)$, find an integer $q \geq 1$ and a sequence of integers $0 = l_0 < l_1 < \cdots < l_{q-1} < l_q = n$ such that the total cost $TC = \sum_{i=0}^{q-1} C(l_i, l_{i+1})$ is minimized.*

### 2.1.1 A Standard Algorithm (Wilber, 1988)

To gain some intuition about how to solve the least-weight subsequence problem, we first outline an algorithm with an $O(n^2)$ time, $O(n)$ memory requirement. To this end we introduce the auxiliary variables $m_{ij}$, which is the sum of the cost obtained after solving the LWSS problem for the points $x_1 \ldots x_i$ plus the cost of $C(i, j) := C(\{\!\{x_{i+1}, \ldots, x_j\}\!\})$. We call $m_{ij}$ the *conditional minimal cost*. More specifically, $m_{0j} := C(0, j)$ is simply the cost of considering the first $j$ points to be in a single cluster. The remaining conditional minimal costs are then recursively defined as $m_{ij} := \min_{l<i} m_{li} + C(i, j)$. We notice that $\min_{l<i} m_{li}$ is the minimal total cost required to solve the least-weight subsequence problem on the points $x_1 \ldots x_i$, and so $m_{ij}$ can be interpreted as the sum of the costs of (i) assigning the first $i$ points optimally and (ii) assigning the points $\{\!\{x_{i+1}, \ldots, x_j\}\!\}$ to a single cluster.

To efficiently compute all $m_{ij}$ we make use of a lookup array $A$ which we fill with the relevant $m_{ij}$ entries as follows: We first compute $m_{01}$ and store it in array $A$ at position 1. Next, we compute the conditional minimal costs $m_{02}$ and $m_{12}$. To compute $m_{12}$ we read the entry $m_{01}$ from our lookup array $A$ at position 1. We store the minimum of $m_{02}$ and $m_{12}$ in array $A$ at position two. We can continue in the same way iteratively. In each iteration we increase $j$ by one and compute all the conditional minimal costs $m_{ij}$ with $i < j$. Each $m_{ij}$ requires the value $\min_{l<i} m_{l,i}$ which we can read from our array $A$ at position $i$. At the end of each such iteration we store $\min_i m_{ij}$ in our array $A$ at position $j$ such that it is available for further computations. The algorithm terminates once we have found $\min_i m_{in}$ which corresponds to the minimal total cost. Overall, this takes $O(n^2)$ time if we can compute the cost function $C$ in $O(1)$.

In univariate microaggregation we are usually more interested in a minimal cost clustering rather than the minimal cost itself. This can be achieved by slight adaptation of the above algorithm. Whenever we compute $\min_i m_{ij}$ we also store the index $i = \arg\min_i m_{ij}$ of the minimum in a linear array $A_{\mathrm{ind}}$ at position $j$. The full pseudo code for this algorithm can be found in Alg. 2. The result of this standard algorithm implicitly holds the information about a minimal cost clustering which can be obtained by *backtracking* in $O(n)$. Starting with $j = n$ we consider $i_{\min} = \arg\min_i m_{ij}$ which we can read from our array $A_{\mathrm{ind}}$ at position $j$. All points with index $l$ with $i_{\min} < l \leq j$ belong to one cluster in the optimal costs clustering. If $i_{\min} > 0$ we repeat the same process with $j = i_{\min}$ to obtain the next cluster. Full pseudo code of this backtracking procedure is provided in Alg. 3 in the appendix.

### 2.1.2 Algorithms for Concave Cost Functions

More efficient algorithms than the standard algorithm outlined above are known for the *concave least weight subsequence* problem. The problem is called concave if the cost function $C$ satisfies for all $0 \leq a < b < c < d \leq n$:

$$C(a, c) + C(b, d) \leq C(a, d) + C(b, c). \tag{1}$$

The above constraint, is sometimes called "quadrangle inequality". If a cost function fulfills this requirement it is called *concave* Wilber (1988); Galil & Park (1990).

If we consider non negative cost functions (i.e. $\forall_X C(X) \geq 0$) with the property that a one-element cluster has no cost ($C(i, i+1) = 0$), we can derive a few interesting implications about concave cost functions. The first observation is, that *wide is worse*. Mathematically this means that for $a < b < c$

$$C(a, b) \leq C(a, c)$$
$$C(b, c) \leq C(a, c).$$

In simple words, this means that if we widen a cluster by adding more points to the left or to the right the cost does not decrease. A similar consequence of concavity is that *splitting is beneficial*, i.e. for all $a < b < c$

$$C(a, b) + C(b, c) \leq C(a, c). \tag{2}$$

Note that the above equation is effectively an opposite of the triangle inequality (which would hold for so-called convex cost functions).

Most important for solving least weight subsequence problems is the consequence that the associated matrix $\mathbf{C}$ with entries $\mathbf{C}_{i,j} = C(i, j)$ is a *Monge matrix* (where for notational convenience we index the rows starting from zero and the columns starting from 1). Most relevant for our purposes is that Monge matrices and their transpose are *totally monotone* (Burkard et al., 1996), i.e., it holds for each $2 \times 2$ submatrix

$$\begin{pmatrix} \alpha & \beta \\ \gamma & \delta \end{pmatrix}$$

that $\gamma < \delta$ implies that $\alpha < \beta$ and, similarly, we have $\gamma = \delta \Rightarrow \alpha \leq \beta$. It is easy to see that if we add a constant value to a column of a totally monotone matrix, the resulting matrix is again totally monotone. This implies that the transposed conditional minimal cost matrix $\mathbf{M}^T$ with entries $\mathbf{M}_{ij} = m_{ij}$ is totally monotone, as it is obtained from the transposed cost matrix $\mathbf{C}^T$ (which is Monge) by adding constant values ($\min_l m_{li}$) to the columns. Finding all minima within the rows of implicitly defined totally monotone matrices of shape $n \times m$ ($n \leq m$) is possible in $O(m)$ using the SMAWK algorithm (Aggarwal et al., 1987).

Using the SMAWK algorithm, dynamic programming (Wilber, 1988; Galil & Park, 1990) allows us to solve the concave least weight subsequence problem in $O(n)$ if the concave cost function $C$ can be computed in $O(1)$ time using $O(n)$ time for preprocessing. While the works (Wilber, 1988; Galil & Park, 1990) focus on concave cost functions $C$, we notice that the presented algorithms in (Wilber, 1988; Galil & Park, 1990) will also work if the corresponding transposed cluster cost matrix $\mathbf{M}^T$ is only *totally monotone*. As we will see in the next chapter, univariate microaggregation for concave cost functions can be phrased as a least weight subsequence problem with an non-concave adapted cost function $C_{\text{adapt}}$. The corresponding transposed cluster cost matrix $\mathbf{M}^T$ is nonetheless totally monotone which allows for the algorithms from (Wilber, 1988; Galil & Park, 1990) to be applied to univariate microaggregation.

## 3 Faster Univariate Microaggregation

In the following we present what properties of the cost function can be used to efficiently solve the univariate microaggregation problem. We first reformulate UM as a least weight subsequence problem (LWSS) which allows for $O(n^2)$ algorithms. We call cost functions which allow this reformulation *ordered minimizable*. If cost functions additionally also have the *splitting is beneficial* property, $O(kn)$ algorithms are possible. For the very common type of *concave cost functions*, we show that UM can be solved in $O(n)$. Lastly, we present an $O(n)$ algorithm, the staggered algorithm, which shows faster empirical running for the concave UM problem compared to more generic $O(n)$ LWSS algorithms presented in (Wilber, 1988; Galil & Park, 1990).

### 3.1 Reformulating Univariate Microaggregation as a Least Weight Subsequence Problem

When comparing UM and LWSS, the most apparent issue is, that in the LWSS problem the values are inherently ordered while for UM no such inherent order is apparent for all cost functions. We will thus consider properties of cost functions that imply an inherent order. We will now characterize those cost

functions that allow a reformulation of the univariate microaggregation problem as a least weight subsequence problem. Parts of the below work is inspired by the treatment of SSE by Domingo-Ferrer & Mateo-Sanz (2002).

**Definition 3** (Pairwise ordered sets). *We call two (multi-)sets $A$ and $B$ pairwise ordered iff all elements in one set are smaller or equal than those in the other, i.e. $\forall_{a \in A, b \in B}\, a \leq b$ or $\forall_{a \in A, b \in B}\, b \leq a$. We call a partition pairwise ordered iff all pairs of multi-sets in the partition are pairwise ordered.*

Equipped with this definition we can characterize those cost functions whose UM problem can be reformulated as a least weight subsequence problem.

**Definition 4** (Ordered minimizable). *We call a total cost $TC$ ordered minimizable if it is sufficient to consider only pairwise ordered partitions when minimizing the total cost $TC$ over all relevant[1] partitions of the universe $\Omega$.*

To grasp the importance of definition 4, lets consider the case of minimizing an ordered minimizable total cost for a bipartition. If there are $n$ elements there are $n-1$ pairwise ordered partitions splitting the data in two parts, but there are $2^n - 2$ partitions in total. So being able to *only* check all pairwise ordered partitions makes a huge computational difference.

While Definition 4 is probably the least restrictive definition, most widely used cost functions fulfill the more restrictive following definition which mixes both the UM and k-means restrictions.

**Definition 5** (q-partition ordered minimizable). *A total cost function $TC$ is q-partition ordered minimizable iff for all partitions $P = \{\!\{X_1, \ldots, X_q\}\!\}$ with multi-sets of cardinalities $S_P := \{\!\{|X_i|\ for\ i \in \{1, \ldots, q\}\}\!\}$ there is a pairwise ordered partition $P'$ consisting of multi-set with cardinalities $S_{P'} = S_P$ that fulfills $TC(P') \leq TC(P)$.*

**Theorem 6.** *If a total cost function is 2-partition ordered minimizable, then it is ordered minimizable.*

In the proof of Theorem 6 we show that if a cost function is 2-partition ordered minimizable it is $q$-partition ordered minimizable for all $q$. This implies that it is ordered minimizable. For the full proof see Appendix B.5.1 in the appendix. Using Theorem 6, it can be shown that the following cost functions are ordered minimizable.

**Corollary 7.** *The total costs $TC = \sum C(X)$ associated to following cost functions $C$ are ordered minimizable:*

- *Sum of Squares Error $C(X) = SSE(X) = \sum_{x \in X}(x - \bar{X})^2$*

- *Sum of Absolute Error $C(X) = SAE(X) = \sum_{x \in X}|x - M(X)|$, where $M$ is the median of $X$*

- *Maximum distance $C(X) = C_\infty(X) = \max_{x \in X}|x - \overset{\infty}{X}|$, where $\overset{\infty}{X} = (\max(X) + \min(X))/2$*

- *Round up error $C(X) = C_\uparrow(X) = \sum_{x \in X}|x - \max(X)|$*

- *Round down error $C(X) = C_\downarrow(X) = \sum_{x \in X}|x - \min(X)|$*

For the proof of Corollary 7 see Appendices B.5.3 to B.5.6. The sum of absolute error cost has been previously proposed under the name absolute deviation from median (ADM) (Kabir & Wang, 2011). For the scenario in which all values $x_i$ are integers, a sum-of-squares distortion metric with rounded mean was also proposed (Mortazavi, 2020) to be able to perform all operations on integers only.

When dealing with any ordered minimizable cost function, we thus can sort the input data either in ascending or descending order (ascending is usually preferred for numerical reasons). Depending on the data, sorting can be done in $O(n \log n)$ or $O(n)$ (e.g. radix-sort on small integers). Currently, sorting is part of any practical algorithm to solve UM exactly and we thus consider the time complexity of all algorithms on sorted data, thereby neglecting the potential additional cost of $O(n \log n)$.

---

[1]For UM the relevant partitions are those which respect the minimum cluster size constraint. For k-means the relevant partitions are those of size $k$.

If cost functions are 1) ordered minimizable, and 2) can be computed in $O(1)$ based on an $O(n)$ preprocessed data, we solve UM in $O(n^2)$ time and $O(n)$ space. This can be done by adapting the cost function of the classic algorithm for LWSS problems such that large cluster costs are returned if the cluster size is smaller than $k$:

$$C_{\text{adapt}}(i,j) = \begin{cases} \text{VAL} & j - i < k, \\ C(i,j) & \text{else.} \end{cases} \tag{3}$$

Here VAL is an arbitrary large enough number, such that the cluster cost $C$ is always smaller than VAL, i.e., $C(i,j) < \text{VAL}$ for all valid choices $i < j$. The following lemma ensures that we can use this adaptation for all previously considered cost functions.

**Lemma 8.** *The cost functions introduced in corollary 7 can be computed in $O(1)$ using $O(n)$ time for preprocessing and consuming no more than $O(n)$ additional memory.*

For the proof of Lemma 8 see Appendix B.3.

Many cost functions are even more structured, which allows faster algorithms as we will see in the next subsection.

### 3.2 Univariate Microaggregation for Cost Functions where Splitting is Beneficial

If the ordered minimizable cost function also has the splitting is beneficial property (i.e., eq. 2 holds), we can improve upon the $O(n^2)$ time complexity for the classical algorithm. The main insight is that we do not need to consider clusters that contain less than $k$ or more than $2k - 1$ points. We can "forbid" these clusterings by adapting the normal cluster cost. For too small/too large clusters we return large costs which ensure the forbidden assignments are never selected. Thus we adapt any cost function with the splitting is beneficial property as

$$C_{\text{adapt}}(i,j) = \begin{cases} \text{VAL} & j - i < k, \\ \text{VAL} & j - i \geq 2k, \\ C(i,j) & \text{else.} \end{cases} \tag{4}$$

Similar to LWSS we can define a matrix $\tilde{\mathbf{M}}$ with entries $\tilde{\mathbf{M}}_{i,j} = \min_{l < i} m_{li} + C_{\text{adapt}}(i,j)$. If we consider the structure of the $\tilde{\mathbf{M}}$ matrix (see colors in Figure 3), we find that there are only $k$ entries per column that need to be considered (green entries in Figure 3). In particular, when computing the minimum value in each column of the matrix $\tilde{\mathbf{M}}$ only entries $\tilde{\mathbf{M}}_{ij}$ with $j - 2k + 1 \leq i \leq j - k$ are necessary to be computed to find the minima of column $j$. This allows a complexity improvement of the classic algorithm from $O(n^2)$ to $O(kn)$. We conclude this subsection by noting the introduced considerations are applicable to the presented cost functions.

**Corollary 9.** *All the costs presented in Corollary 7 have the splitting is beneficial property.*

This is true as they are concave (see next subsection, Lemma 10), they have $C \geq 0$ and $C(i, i + 1) = 0$.

### 3.3 Univariate Microaggregation for Concave Cost Functions

Besides the classical algorithm used to solve least weight subsequence problems, more advanced algorithms (Wilber, 1988; Galil & Park, 1990) are available to solve *concave* least weight subsequence problems in $O(n)$. We observe that those algorithms will also work if the associated cluster cost matrix $\mathbf{M}^T$ is only totally monotone. The previous definition of $C_{\text{adapt}}$ (eq. (3)) needs to be slightly modified such that the matrix $\tilde{\mathbf{M}}^T$ is totally monotone for these algorithms to be applicable.

$$C_{\text{adapt}}(i,j) = \begin{cases} \text{VAL}(i) & j - i < k, \\ C(i,j) & \text{else.} \end{cases} \tag{5}$$

We again choose the large costs $\text{VAL}(l)$ large enough such that $C(i,j) < \text{VAL}(l)$ for any $l$ ($1 \leq l \leq n$) and any valid $i$, $j$. Further, we require $\text{VAL}(l) < \text{VAL}(m)$ for $l < m$ to ensure total monotonicity of the transposed cluster cost matrix $\tilde{\mathbf{M}}^T$.

If our cost function is concave this definition of the adapted cost $C_{\text{adapt}}$ ensures that, the *transposed* cluster cost matrix $\tilde{\mathbf{M}}^T$ is totally monotone (see appendix for a proof, a visualization of such a matrix is provided in Figure 3 if we assume the orange entries are also green). For cost functions $C$ in that can be evaluated in $O(1)$ time (for $O(n)$ preprocessed data) we can use $C_{\text{adapt}}$ as cost function and solve this problem using dynamic programming algorithms, e.g., via the algorithm by Wilber (1988), or an algorithm proposed by Galil & Park (1990), to obtain a solution for univariate microaggregation on sorted input in $O(n)$ time and space.

For the special case of the sum of squares cost, it has been previously shown, that the cost is concave (Grønlund et al., 2018; Wu, 1991) and can be computed in $O(1)$ (using $O(n)$ time and space for preprocessing) using a cumulative sum approach. These results have been used to speed up the 1D k-means problem (Grønlund et al., 2018), but have not yet been used to solve univariate microaggregation tasks. Let us assert that all of the considered cost functions in Corollary 7 are indeed concave:

**Lemma 10.** *The cost functions from Corollary 7 are concave.*

For the proof of Lemma 10 we refer to the appendix Appendix B.4.

Putting together Corollary 7, Lemma 8, and Lemma 10 we obtain the following result:

**Theorem 11.** *On sorted data, we can compute optimal univariate microaggregation in $O(n)$ time and space for the cost functions introduced in Corollary 7.*

Naturally, Theorem 11 extends to unsorted data if we pay at most an additional $O(n \log n)$ cost for sorting.

## 3.4 Univariate Microaggregation for Concave Costs where Splitting is Beneficial

If a cost function is concave and splitting is beneficial we can combine restricting the cluster size and preserving total monotonicity to arrive at the following adapted cost

$$C_{\text{adapt}}(i,j) = \begin{cases} \text{VAL}(i) & j - i < k \\ \text{VAL}(-i) & j - i \geq 2k \\ C(i,j) & \text{else} \end{cases} \tag{6}$$

As before $\text{VAL}(l)$ should be large enough such that $C(i,j) < \text{VAL}(l)$ for all valid $i$, $j$, but this time not only for positive $l$, but also for negative $l$ with values beteen $-n \leq l \leq n$. We require again that $\text{VAL}(l) < \text{VAL}(m)$ for $l < m$ to ensure total monotonicity of the transposed cluster cost matrix $\tilde{\mathbf{M}}^T$. We proove that for this definition of the adapted cost $C_{\text{adapt}}$ the *transposed* cluster cost matrix $\tilde{\mathbf{M}}^T$ is totally monotone in appendix Appendix B.2.2. The structure of the cluster cost matrix correspond to this definition of $C_{\text{adapt}}$ is displayed in Figure 3.

When a cost function has both the splitting is beneficial property and is concave, it is possible to significantly decrease the runtime of the classical algorithm introduced in Section 3.2. To this end it is instrumental to observe that in totally monotone matrices, the positions of the minima of the rows are non decreasing (see Alg. 2 for pseudo-code). Although this does not provide an asymptotic runtime improvement, it greatly decreases empirical runtime for larger $k$ (see Figure 4).

By incorporating the cluster size constraints, it is moreover possible to improve the generic algorithms for least weight subsequence problems (Wilber, 1988; Galil & Park, 1990) for concave cost functions with the splitting is beneficial property. We present an algorithm for this problem in the next subsection.

## 3.5 The Staggered Algorithm

When solving UM, the algorithm by Wilber (Wilber, 1988) and the algorithm by Galil and Park (Galil & Park, 1990) are both generic dynamic programming algorithms that work with any kind of ordered minimizable concave cost function. They achieve the linear running time by repeatedly calling the SMAWK

algorithm Aggarwal et al. (1987) to compute the minima of rows of submatrices of the totally monotone matrix $\mathbf{M}^T$. These algorithm only use the concavity and are oblivious to the restriction on the cluster size and the resulting structure of the $\tilde{\mathbf{M}}^T$ matrix (compare green entries in Figure 3). In contrast, we have the $O(kn)$ algorithm (Domingo-Ferrer et al., 2008) or the adapted classical algorithm (Section 3.2) which are unaware of the concavity of the cost function and use the restrictions on cluster sizes to achieve speed. By using both the restrictions on the cluster sizes and the concavity of the cost function we can obtain an algorithm that empirically runs faster than the other $O(n)$ algorithm on UM tasks, and the asymptotic runtime is still $O(n)$. We present the pseudo code for one such algorithm, "the staggered algorithm", in Alg. 1. The main idea of the staggered algorithm is to apply the SMAWK algorithm to matrices along the diagonals that correspond to valid cluster assignments (see Figure 3 for a visualization of this idea). For UM tasks, the algorithm is empirically faster as fewer calls of $C_{\text{adapt}}(i,j)$ are wasted on combinations of $i,j$ which are forbidden by cluster size constraints. Lastly, the staggered algorithm is conceptually simpler than the other two $O(n)$ algorithms which employ a guessing and recovery strategy to cope with unavailable entries $\min_{l<i} m_{li}$. For the staggered algorithm, no such guessing needs to be employed, as all necessary entries are available when making calls to the SMAWK algorithm.

This algorithm has space and runtime complexity of $O(n)$ for cost functions that can be computed in $O(1)$ using $O(n)$ preprocessing. To see this, not that any call to the SMAWK algorithm is $O(k)$ and there at most $\lceil \frac{n}{k} \rceil$ calls to the SMAWK algorithm. This means that all the calls to the SMAWK algorithm are at most $O(n)$.

**Input:** sorted array $v$, minimum size $k$, cost function calculator $C$
**Output:** array implictly representing the optimal univariate microaggregation of v
**1** let $n = \text{length}(v)$
**2** **if** $n \leq 2k - 1$ **then**
**3** $\quad$ **return** all zeros array of length $n$
**4** C.do_preprocessing(v, k)
**5** SMAWK.cost = C
**6** **for** $i \in \{k, \ldots, 2k - 1\}$ **do**
**7** $\quad$ SMAWK.MinTotalCost[i] = C.calc(0, i)
**8** $\quad$ SMAWK.ArgminTotalCost[i] = 0
**9** SMAWK.col_min$(2k, \min(3k - 1, n), k, 2k - 1)$
**10** **if** $n \leq 3k - 1$ **then**
**11** $\quad$ **return** SMAWK.ArgminTotalCost
**12** $f, R = (n - 3k + 1) \operatorname{divmod} k$
**13** **for** $i \in \{3, \ldots, 2 + f\}$ **do**
**14** $\quad$ SMAWK.col_min$(ik, (i + 1)k - 1, (i - 2)k + 1, ik - 1)$
**15** **if** $R > 0$ **then**
**16** $\quad$ $i = 3 + f$
**17** $\quad$ SMAWK.col_min$(ik, n, (i - 2)k + 1, n - 1)$
**18** **return** SMAWK.ArgminTotalCost

**Algorithm 1:** Pseudo code for the staggered algorithm. This algorithm uses both the concavity of the cost function through the use of the SMAWK algorithm and the restrictions, that clusters are of size at least $k$ and at most $2k - 1$. When calling SMAWK.col_min(i,j,k,l), the SMAWK algorithm is applied to compute the column minima of a submatrix of the implicitly defined cost matrix $\mathbf{M}$ which contains columns $i$ through $j$ and rows $k$ to $l$ (see boxes in Figure 3). After executing the SMAWK algorithm, the minimal total cost for each column is stored in MinTotalCost and the row that corresponds to that value is stored in ArgminTotalCost.

Figure 3: Visualization of how the staggered algorithm processes the cluster cost matrix $\tilde{M}$ for a cost function that is concave and has the splitting is beneficial. The matrix is shown for $n = 8$ and $k = 2$. Red entries correspond to clusterings which contain clusters that are to small. Orange entries correspond to entries that contain unnecessarily large clusters and green entries need to be processed to find an optimal univariate microaggregation clustering. In Alg. 1 lines 5-7 initialize $m_{02}$ and $m_{03}$. Line 9 processes the first submatrix indicated by the square overlay in columns 4 and 5. For most inputs the loop in lines 12 & 13 processes most of the matrix (here only columns 6 & 7). Lastly, potentially remaining columns are processed in line 15-17.

## 4   Univariate Microaggregation on real hardware

So far we have considered the mathematical foundations underlying fast UM algorithms. In practice though, the computations involved are not happening at arbitrary precision. On most current hardware, calculations involving reals are executed with fixed width floating point numbers (e.g. 32 or 64 bits). We notice that utilizing the presented algorithms/cost functions can result in suboptimal clusterings, due to finite precision of these floating point numbers. We thus dedicate the next two subsections to the analysis and mitigation of errors caused by finite precision floating point numbers. At the end of the section we also highlight a real world runtime improvement possible for some of the cost functions. As a running example we will consider the sum of squares cost function, though many of the considerations carry over to other cost functions as well.

The default approach of computing the $SSE$ in $O(1)$ is to first precompute (in $O(n)$) the cumulative sums

$$s^{(1)}(j) = \sum_{i \leq j} x_i \quad \text{and} \quad s^{(2)}(j) = \sum_{i \leq j} x_i^2$$

with $s^{(1)}(0) = s^{(2)}(0) = 0$. Then we can compute for $i < j$

$$SSE(i,j) = s^{(2)}(j) - s^{(2)}(i) - \frac{\left(s^{(1)}(j) - s^{(1)}(i)\right)^2}{j - i} \tag{7}$$

which is constant time irrespective of the values of $i$ and $j$.

### 4.1   Floating Point Errors During Preprocessing

When computing the cumulative sum during preprocessing, the running sum can grow and reach quantities which are no longer represented well numerically. This can lead to erroneous cluster cost being calculated which in turn may lead to suboptimal clusterings. As an example, computing the sum of squares error according to eq. (7) on the integers (i.e. the universe is $\Omega = \{0, 1, 2, \dots\}$) using 64bit floats returns the wrong result $SSE(300\,079, 300\,082) = 1$ (which should be 2). We can do better than the simple cumulative sum approach, by observing that we only need to consider $C(i,j)$ with $j - i \leq 2k - 1$ for UM. This means we don't need the full cumulative sum and can get away with sums of fewer elements.

We therefore consider partial cumulative sums which are cumulative sums which we restart every $k$-th time. So in preprocessing we compute the partial cumulative sums

$$s_{i,j}^{(1)} = \sum_{i < l \leq j} x_l$$

for $i = 0, k, 2k, \ldots$ and $j(i) = i + 1, i + 2, \ldots, i + k$, and we set $s_{i,i}^{(1)} = 0$. We can do this similarly for the cumulative sums of the squared entries. Both of these preprocessing steps take $O(n)$ time and $O(n)$ space.

Using these partial cumulative sums, we can also compute the sum of squares error for univariate microaggregation. Let us denote with $m, r = i \mod j$ the modulus $m$ and remainder $r$ of the integer division of $i$ by $j$. Then we can compute the SSE from the $s_{i,j}$ for a univariate microaggregation problem with min size $k$: For $i < j$ compute $m_i, r_i = i \mod k$ and $m_j, r_j = j \mod k$. We now notice that for valid entries $i, j$ we have $m_j - m_i \leq 1$ as $j - i \leq 2k - 1$. Lets first consider $m_j = m_i$, then we compute the SSE as

$$SSE(i,j) = s_{m_j,r_j}^{(2)} - s_{m_j,r_i}^{(2)} - \left( s_{m_j,r_j}^{(1)} - s_{m_j,r_i}^{(1)} \right)^2 / (j - i).$$

In the case that $m_j = m_i + 1$ we compute the SSE as

$$SSE(i,j) = s_{m_j,r_j}^{(2)} + s_{m_i,k}^{(2)} - s_{m_i,r_i}^{(2)} - \left( s_{m_j,r_j}^{(1)} + s_{m_i,k}^{(1)} - s_{m_j,r_i}^{(1)} \right)^2 / (j - i).$$

Both of these are $O(1)$ but the numbers involved in the cumulative sums are much smaller which can decrease floating point representation errors. This consideration is particularly important for the case of $SSE$ and $MAE$ both of which use cumulative sums in their computations.

## 4.2 Floating Point Errors when Computing the Total Cost Function

When computing the total cost function $TC$ we are computing the sum of at least $\lfloor \frac{n}{2k-1} \rfloor$ values with equal sign, thus $TC$ is increasing in magnitude with each summand. Yet, floating point numbers have a higher absolute resolution close to zero. It would thus be numerically advantageous if we could keep the total cost $TC$ close to zero. For cost function with the splitting is beneficial property this can be achieved without increasing the runtime complexity by regularly resetting the stored $TC$ values. Most algorithms work by first computing and storing the minimal total cluster cost $TC_{\min}(q) := \min_i m_{iq}$ of the first $q$ points (see Section 2.1 for the definition of $m_{ij}$). Then the algorithms find the optimal clustering of the next $q + p$ points based on the optimal clustering of those previous values up to $q$. We notice that $TC_{\min}(l)$ increases as $l$ increases but for cost functions where splitting is beneficial we only need to know the last $2k - 1$ total cost values, i.e., we need to know $TC_{\min}(q)$ for $q \geq l - (2k - 1)$ to compute the total cost $TC_{\min}(l)$. Thus for certain algorithms we can reduce the growth of the total cost as follows. On a high level, we subtract the smallest stored and still needed minimal total cost from the other stored and still needed minimal total cost values at regular intervals. As an example, consider the classical algorithms simple and simple+ with their pseudocode in Alg. 2. To implement the strategy, we could expand the loop spanning lines 7-9 in Alg. 2:

**1** [continuing the loop in lines 7-9 in Alg. 2]
**2** Let $i_{\min}$ be the lower bound of the argmin in line 8 of Alg. 2
**3** MinNeededMinCost $= \min_{i_{\min}+1 \leq i \leq j}$ MinCost[i]
**4** **for** $i \in \{i_{\min} + 1, \ldots, j\}$ **do**
**5** $\quad \lfloor$ MinCost[j] = MinCost[j] - MinNeededMinCost

Here we subtract the still needed minimal total cost MinNeededMinCost from those MinCost values that are still needed in future iterations. The values that are still needed in future iterations of the main loop are designated mostly by the lower bound $i_{\min}$. This lower bound differs depending on whether we consider the simple or simple+ algorithm. It is possible to do a similar procedure for the staggered algorithm without changing the runtime complexity of either algorithm. Using such a procedure, we can ensure that the total cost $TC$ doesn't grow as a function of $\approx \frac{n}{2k-1}$ but remains nearly constant (in the case of simple/simple+) or growth as a function $\propto k$ in the case of the staggered algorithm. Thus for small $k$ and large $n$ the described procedure allows us to make use of the higher absolute floating point resolution near zero.

## 4.3 Faster Cost Function Evaluations

As a final consideration for implementing UM on real hardware, we consider how to compute the cost function with fewer instructions. All the here presented algorithms work by computing the optimal partitions of the

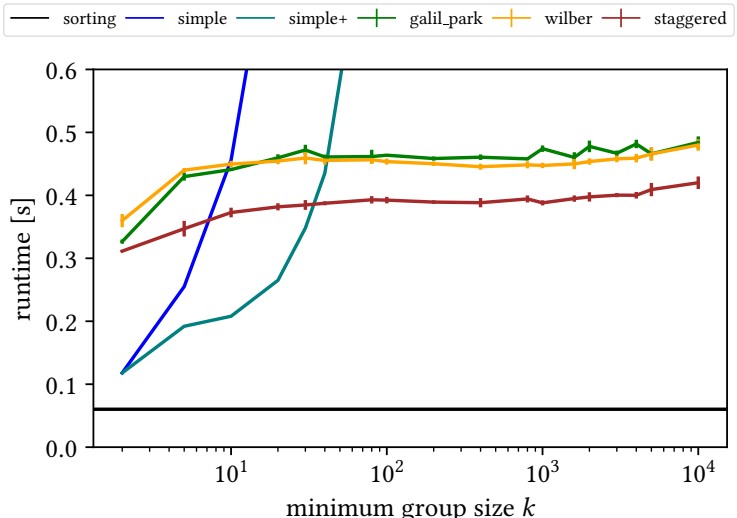

Figure 4: Runtime on random data of size 1 million as a function of minimum group size $k$. Colors indicate different algorithms as displayed in the legend above. The time to sort the elements is included as a baseline (black). Shown are mean and standard deviation over 10 repeats. As cost function, the sum-of-squares cost function as introduced in section 4.1 is used. The simple dynamic program (denoted as "simple", pseudocode in Alg. 2) is representative of algorithms previously proposed (e.g. Domingo-Ferrer & Mateo-Sanz 2002). It is slower than the newly proposed dynamic program (simple+, pseudocode in Alg. 2) for any minimum group size $k$. For small values of $k$, the $O(kn)$ dynamic programs simple and simple+ are faster than the $O(n)$ algorithms staggered, Wilber, and Galil Park. For large values of the minimum group size $k \geq 30$ the $O(n)$ dynamic programs are faster than even the improved $O(kn)$ program.

first $q$ elements of the universe $\Omega$. Let us denote these first $q$ elements with $\Omega[\ldots, q]$. From the optimal partition of the first $q$ points, the optimal partition of the first $q + 1$ points of the universe can then be computed. During this process we only compare the total cost of partitions of the same set $\Omega[\ldots, q]$. Stated differently, we compare the total cost $TC(P_{\Omega[\ldots,q]})$ of one partition $P_{\Omega[\ldots,q]}$ of the first points $\Omega[\ldots, q]$ with the total cost of another partition $P'_{\Omega[\ldots,q]}$ of the same set $\Omega[\ldots, q]$. If we are only interested in the resulting clustering and not in the actual cost of the clustering, we can minimize cost functions which require fewer operations to compute than the original cost functions. This idea is more easily understood with an example. Lets assume we want to minimize the $SSE$ on partitions of the first $q$ elements of the universe $\Omega$ then for two partitions $P_{\Omega[\ldots,q]}$ and $P'_{\Omega[\ldots,q]}$ of this set,

$$TC_{SSE}(P_{\Omega[\ldots,q]}) \leq TC_{SSE}(P'_{\Omega[\ldots,q]})$$

$$\Leftrightarrow \sum_{x \in \Omega[\ldots,q]} x^2 - \sum_{X \in P_1} \frac{\bar{X}^2}{|X|} \leq \sum_{x \in \Omega[\ldots,q]} x^2 - \sum_{X \in P_2} \frac{\bar{X}^2}{|X|}$$

$$\Leftrightarrow \sum_{X \in P_1} \tilde{C}_{SSE}(X) \leq \sum_{X \in P_2} \tilde{C}_{SSE}(X)$$

with

$$\tilde{C}_{SSE}(X) = -\frac{\bar{X}^2}{|X|} \tag{8}$$

Overall, we find the clustering that minimizes SSE is equivalent to finding the clustering minimizing $\tilde{C}_{SSE}$ if we only use algorithms that compare clusterings of the same set $\Omega[\ldots, q]$. We can find similar expressions

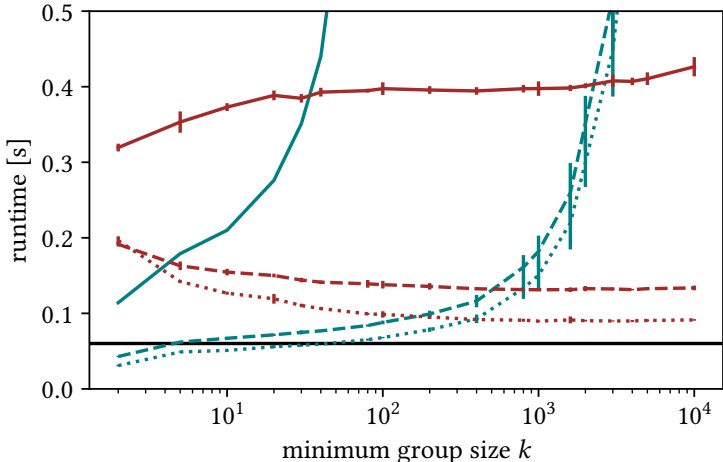

Figure 5: Runtime impact of different methods to compute the cost functions. Colors indicate algorithms, linestyles indicate methods to compute the cost functions. Shown are mean and standard deviation aggregated over 10 runs. The colors red/blue indicate the staggered/simple+ algorithm respectively (same colors as Figure 4). The dashed lines are the default cumulative sum approach (eq. 7), the solid lines are the partial cumulative sum method introduced in section 4.1, and the dotted lines indicate the alternative cost function approach introduced in section 4.3. We see that overall for the same algorithm the alternative cost function is fastest with the default cumulative sum approach being not much slower. The partial cumulative sum approach comes with a big runtime penalty.

for some cost functions, as well:

$$\tilde{C}_\uparrow(X) = |X| \max(X)$$
$$\tilde{C}_\downarrow(X) = -|X| \min(X)$$

## 5 Experiments

In the following we show to kinds of experiments. Firstly, we perform runtime experiments to evaluate whether the theoretical considerations made lead to performance improvements on real hardware. In the second set of experiments we evaluate whether projection based approaches are a possible competitor to solve the multivariate microaggregation task.

### 5.1 Runtime Experiments

We compare the algorithms by Galil and Park, Wilbers algorithm, and two versions of a simple dynamic program, one which does only use the restrictions on cluster sizes (simple) and another one which additionally uses that the row minima of the $M^T$ matrix are non decreasing (simple+). All the methods were implemented in python and compiled with the numba compiler. We additionally include the time to sort the initially unsorted data of the same size for comparison. We generate our synthetic data set by sampling one million reals uniformly at random between zero and one.

For low values of the minimum group size $k$ the simple dynamic programs are faster than the $O(n)$ algorithms, with the simple+ algorithm outperforming the simple algorithm in terms of computation time. These simple algorithm become slower than the more complex dynamic programs when the minimum group size $k$ exceeds 250. When comparing the $O(n)$ algorithms, the staggered algorithm is overall about 20 percent faster than other algorithms. Overall, we see that the algorithms including both the concavity of the cost function and the minimum / maximum group size constraints, i.e., the simple+ and staggered algorithm, are faster than those algorithms that don't include both constraints.

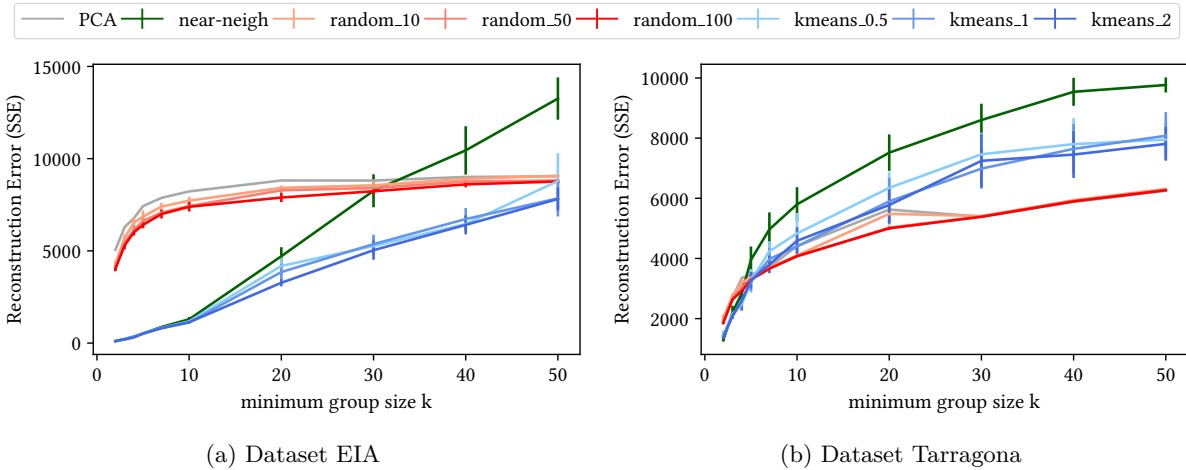

(a) Dataset EIA

(b) Dataset Tarragona

Figure 6: Reconstruction Error for the Multivariate Microaggregation task on real world datasets. We show the Reconstruction Error or loss (lower is better) for the SSE cost function for different values of minimum group size $k$. We compare four different approaches, two projection based approaches 'PCA' and 'random' projection as well as two approaches designed for the multivariate case the 'kmeans' based approach and a 'nearest-neigh' based approach by (Domingo-Ferrer et al., 2008). For the PCA and random approach, we use a vector to project the multivariate task into one dimension and solve the univariate microaggregation task with the methods introduced here. For the random projection approach one run consist of making 10, 50, and 100 random projections (as indicated in the legend) and taking their minimum. For the $k$-means + cleanup based approach, we first performed k-means where the number of clusters $k$ is $f$ dataset_size/$k$ where $f$ is a factor of 0.5, 1 or 2 as indicated in the legend. To make sure that the obtained data fulfills the minimum group size constraint we merge clusters of to small size with the closest clusters until all clusters are large enough. For the random, nearest-neighbor and $k$-means based approach we show means and standard deviation over 10 runs. On the EIA dataset (a), the k-means based approaches outperform the projection based approaches by a large margin. The nearest-neighbor based approach is performing on par with the k-means based approach for low $k$ but gets increasingly worse with increasing $k$. On the Tarragona dataset (b) the picture is not as clear. For low values of $k \leq 5$ the k-means based approach is better or equally good as the projection based approaches while for larger values of $k$ the projection based approaches show less reconstruction error. The PCA based approach is usually slighlty worse than the random projection based approach. The nearest-neighbor approach on par with the k-means based approach for $k$ less than 5 but it gets increasingly worse with increasing values of $k$.

In Figure 5 we compare the different ways of computing the cost functions exemplary for the simple+ and staggered algorithm. Within the same algorithm we find that the method from section 4.3 is fastest with the default cumulative sum (eq. 7) being a close competitor. The numerically more stable method introduced in section 4.1 is about four times slower than the default method.

## 5.2 Solving Multivariate Microaggregation through projection

In practice one is frequently met with Multivariate Microaggregation problems, i.e. there are multiple descriptors that need to be aggregated simultaneously. In this section we compare approaches designed for the multivariate problem with approaches which work by projecting the multivariate problem into 1d, solving the univariate problem with the methods discussed here and then project the solution back to the multivariate setting.

The most simple approach to project the multivariate data into one dimension is to use principal components analysis (PCA). PCA identifies the axes of maximal variance. We then use the axis with maximal variance to project the multivariate data into a single dimension, we then solve the 1d problem to obtain the cluster assignment. A similar projection based approach uses random projections, i.e. we sample random vectors

with random values between zero and one as our projection vector. To achieve reasonable results, we try multiple random vectors and take the one with lowest multivariate loss. We compare these two projection based approaches with two approaches designed for the multivariate case. The first one works by obtaining a heuristic solution to the k-means problem. Because there might be clusters which are to small, we merge the clusters of size less than $k$ with the closest other cluster (measured by the distance of the centroids) until no cluster of too small size remains. We randomize the order of this merging procedure. As a second method designed for the multivariate case we consider the nearest-neighbor based approach by Domingo-Ferrer et al. (2008). While the original proposed method allows for overlapping clusters[2], i.e. the subsets $X_i$ (compare Definition 1) need not be pairwise disjoint, we slightly modified their procedure such that they are pairwise disjoint for a more fair comparison. We found experimentally that whether we allow for overlapping clusters or not made pretty much no difference in the reconstruction error[3] (the mean reconstruction errors are within 6% of each other, none of the two clearly beating the other and the standard deviations are twice as large).

To evaluate these approaches we use two real world datasets that have been used in previous research on multivariate microaggregation (Domingo-Ferrer & Mateo-Sanz, 2002). The EIA dataset (Brand et al., 2002)[4] consist of 4092 instances and 15 columns, of which we use the 10 numeric columns. The Tarragona dataset (Brand et al., 2002)[5] has 834 instances and 13 columns. We average the results of the $k$-means based approach, the nearest-neighbor based approach, and the random projection approach over 10 runs and show the mean and standard deviation. We measure the quality of the obtained clustering by the multivariate loss (SSE). The results are shown in Figure 6.

We find that on the EIA dataset the k-means based approaches greatly outperform, i.e. have lower reconstruction error, compared to the projection based approaches for almost all values of minimum group size $k$ we considered. It is just that when $k$ approaches 50 the two approaches become more similar in terms of their reconstruction error. The nearest neighbor based approach performs on par with the $k$-means based approach for values of $k$ less than 10. For larger values of $k$ it becomes increasingly worse compared to $k$-means based approach even overtaking the projection based approaches at $k = 30$. On the Tarragona dataset and low values of $k > 5$ the k-means based approaches are winning by a small margin but for larger values of $k$ the projection based approach are performin better than the $k$ means based approaches. When comparing the PCA approach to the random prohjection approach, the random projection approach is usually slightly better than the PCA based approach. From these preliminary experiments, it seems, that the projection based approaches might be a valid competitor on some datasets, while on other datasets, they are not viable. Overall the nearest-neighbor based method performs on par with the $k$-means based approach for small values of $k$ (e.g. less than 10) and is inferior in reconstruction error for larger values of $k$.

## 6    Conclusions

We considered the problem of univariate microaggregation. We characterized properties of cost functions that lead to complexity improvements over the naive algorithm with exponential runtime. By mapping univariate microaggregation to a least weight subsequence problem we showed that UM for ordered minimizable cost functions is solvable in $O(n^2)$. If splitting is beneficial, a maximum group size constraint allows to improve this complexity to $O(kn)$. For ordered minimizable, concave cost functions a different strategy allows to find an optimal solution to univariate microaggregation in $O(n)$ time and space on sorted data. These results apply to many popular cost functions such as sum of squares, sum absolute error, maximum difference as well as round up/down costs. By incorporating both a maximum group size constraint and the concavity of cost functions, we are able to provide an improved $O(kn)$ algorithm (simple+) and an improved $O(n)$

---

[2]The definition of the algorithm is not sound unless this is assumed, even though this is not explicitly mentioned in the original source. In spirit of the presented algorithm we considered the below highlighted overlapping case.

[3]For the overlapping case we assume that there are not only the overlapping sets $X_i$ used to compute the cluster centroids but for each of them a set $Y_i \subset X_i$ exists which highlights the non-auxillary nodes in $X_i$. The $Y_i$ are pairwise disjoint and unify to $\Omega$. We obtain the reconstruction error the following way. For each element we find the only set $Y_i$ it is contained in and compute the (squared) distance to the centroid of the correspond set $X_i$. To obtain the final reconstruction error we sum up those distances for all elements. This naturally captures a slightly different notion of $k$-privacy as the elements of the new dataset $\Omega'$ are obtained by aggregating at least $k$ elements of the original dataset.

[4]https://github.com/sdcTools/sdcMicro/blob/master/data/EIA.rda

[5]https://github.com/sdcTools/sdcMicro/blob/master/data/Tarragona.rda

algorithm (staggered) which demonstrate faster empirical runtime in comparison to other algorithms in their respective complexity class.

*Limitations* In practice one is frequently met with multivariate microaggregation (MM) problems and one of the heuristics used to solve MM problems is to devise an order on the points (see (Mortazavi, 2020) and references therein). The resulting ordered multivariate microaggregation problem is sometimes referred to as a univariate microaggregation problem even though it really is a least-weight subsequence problem with multi-dimensional cost functions. These multi-dimensional cost functions may no longer be concave as the concavity of the cost functions relies on the sorted (i.e. arranged in increasing/decreasing order rather than any fixed but arbitrary order) 1D nature of the points. Thus the presented $O(n)$ algorithms are not applicable to such ordered multivariate microaggregation problems. As the simple $O(kn)$ algorithm does not rely on the concavity of cost functions, it can still be used to solve the ordered multivariate microaggregation problem if a multi dimensional equivalent of the splitting is beneficial property holds.

It is well known, that the privacy concept of k-anonymity may provide insufficient protection if for example the confidential attributes are not sufficiently diverse. Caution should thus be taken if k-anonymity, or microaggregation to achieve k-anonymity, is applied in practice. For practical anonymity guarantees, the concept of differential privacy (Dwork et al., 2006; Dwork, 2008) provides a more robust protection.

While $k$-anonymity relies on the intuition "hidden in a crowd of size at least $k$", the key idea of differential privacy is "plausible deniability", i.e. whether or not any individual is included in the data cannot be said with certainty. To achieve differential privacy guarantees, usually the output of an algorithm is sufficiently perturbed. Li et al. (2012) showed that using $k$-anonymity, it is possible to essentially perform input perturbation and still achieve differential privacy if the $k$-anonymity type algorithm is augmented with two steps. First, the k-anonymity algorithm needs to be $\epsilon$-safe: Intuitively this means the $k$-anonymity algorithm needs to output non-optimal clusterings with nonzero probabilities. Second, $k$-anonymity is applied to a subsample from the original dataset instead of the full dataset (for the full details see (Li et al., 2012)).

*Concave costs where splitting is not beneficial* The majority of cost functions considered are concave and have the splitting is beneficial property which allows all the presented algorithms to be applied. Yet, there are concave cost functions which do not have the splitting is beneficial property. As a simple example consider adding a constant group cost $\delta$ to a concave cost function $C$ which also has the splitting is beneficial property to penalize having too many clusters (i.e., we obtain the adjusted cost function $\hat{C}(i,j) := C(i,j) + \delta$). This adjusted cost function is still concave but no longer has the splitting is beneficial property. Thus the generic $O(n)$ algorithms (Wilber, 1988; Galil & Park, 1990) may still be used to solve UM problems for the modified cost function $\hat{C}$ while the $O(kn)$ algorithms (simple, simple+) and the staggered algorithm are not applicable.

*Relation to k-means* The definitions presented here also allow to have a more fine grained understanding of other 1D clustering problem with different cost functions. As an example let us consider the $k$-means problem which restricts the number of clusters to be exactly $k$. An existing $O(kn^2)$ algorithm (Grønlund et al., 2018) solves the 1D k-means problem if the cost function considered is ordered minimizable. If the cost is additionally also concave, $O(kn)$ and $O(n \log U)^6$ algorithms exist (Grønlund et al., 2018). The splitting is beneficial property, while not explicitly used in algorithms, seems to be implicitly used when formulating the k-means problem. The most popular formulation of the $k$-means problem asks to minimize the clustering cost subject to *exactly* k clusters, while minimizing costs subject to *at most* k clusters could also be thinkable. However, these two formulations will result in identical optimal clusterings if the cost function has the splitting is beneficial property.

*Mixed UM and k-means scenario* Another way to avoid using overly many clusters in a UM setting could be to impose a maximum number of clusters constraint in addition to the usual minimal group size constraint. This would correspond to a mixed UM and k-means scenario. We implicitly showed that the considered cost functions are ordered minimizable even in a mixed UM and k-means scenario, as cost functions that are $q$-partition ordered minimizable also encompass this mixed scenario. Hence, algorithms used to solve the 1D $k$-means problem (see (Grønlund et al., 2018) for many such algorithms) work correctly in the mixed UM and $k$-means scenario if we adapt the cost function as explained in eq. (3).

---

[6]$\log U$ is a number of bits used

*Regularized UM problem* The usual UM problem forbids any clustering with too few entries, which might be overly restrictive when $k$ is larger. In such cases it might be worth considering a regularized UM problem, i.e., we don't entirely forbid invalid clusterings but instead return an additional cost when the minimal cluster size constraint is violated. This can for example be achieved by defining the regularized cluster cost as

$$C_{\text{reg}}(i,j) = C(i,j) + \begin{cases} \lambda\left(k - (j - i)\right) & j - i < k \\ 0 & \text{else} \end{cases}$$

with regularization parameter $\lambda \geq 0$. By adjusting the regularization parameter $\lambda$ to be large we can make it more and more unlikely that the cluster size constraint is violated. If $C$ was concave, the regularized UM problem may also be minimized by the generic concave algorithms (Wilber, 1988; Galil & Park, 1990). Similarly, if splitting was beneficial for $C$, the regularized UM may be solved by a slightly adjusted[7] standard algorithm in $O(kn)$.

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

## A    Additional algorithms

### A.1    Pseudo code for the classic algorithms (simple/ simple+)

**Input:** sorted array $v$, minimum size $k$, cost function calculator $C$
**Output:** array implictly representing the optimal univariate microaggregation of v

**1** let $n = \text{length}(v)$
**2** **if** $n \leq 2k - 1$ **then**
**3**    **return** all zeros array of length $n$
**4** C.do_preprocessing(v, k)
**5** MinCost/Argmin = zero based arrays of size n+1
**6** MinCost[0] = 0
**7** **for** $j \in \{1, \ldots, n\}$ **do**
**8**    Argmin[j] = $\arg\min_{0 \leq i < j}$ MinCost[i] + C.calc(i, j)
**9**    MinCost[j] = MinCost[Argmin[j]] + C.calc(Argmin[j], j)
**10** **return** Argmin[1:n]

**Algorithm 2:** Pseudo code for the **classical algorithm** used to solve least weight subsequence problems. If this algorithm is used to solve univariate microaggregation with ordered minimizable cost functions, the constraint on the arg min in line 8 becomes $0 \leq i < j - k$. If splitting is beneficial the constraint on the arg min in line 8 is $\max(0, j - 2k + 2) \leq i \leq j - k$ (this is the simple algorithm). Lastly, for concave cost functions that also have the splitting is beneficial property, the bound on the arg min is $\max(\text{Argmin[j-1]}, \text{j-2k+2}) \leq i \leq j - k$ with Argmin[0]=0 (this is the simple+ algorithm).

### A.2    Pseudo code for the backtrack algorithm

**Input:** zero indexing based implicit cluster array $b$
**Output:** array explictly representing the clustering

**1** out = zero indexing based array of same length as $b$
**2** p=length($b$); NumClusters=0
**3** **while** *true* **do**
**4**    NumClusters +=1
**5**    **for** $i \in \{b[p-1], \ldots, p-1\}$ **do**
**6**       out[i] = NumClusters
**7**    p=b[p-1]
**8**    **if** $p = 0$ **then**
**9**       break
**10** **for** $i \in \{0, \ldots, length(b) - 1\}$ **do**
**11**    out[i] = NumClusters - out[i]
**12** **return** out

**Algorithm 3:** The **backtrack** algorithm converts an implicit cluster representation array $b$ into an explicit cluster representation. All the presented algorithms return such an implicit representation. For example an implicit cluster array $[0, 0, 1, 1, 2, 2]$ represents the clustering $[0, 1, 2, 2, 2, 2]$ that is the first and second element are their own cluster, and the remaining elements are in one cluster. The backtrack algorithm runs in $O(n)$.

# B    Mathematical details and proofs

## B.1    Multivariate microaggregation is NP-hard for many cost functions

It was previously known that multivariate microaggregation is NP-hard for the sum of squares error cost function. We noticed that the proof provided by Oganian & Domingo-Ferrer (2001) could be easily adapted for other cost functions as well.

Let $l_i(C), i \in \{3, 4, 5\}$ denote the minimal cluster cost for cost function $C$ computed for the structures of size $i$ outlined in (Oganian & Domingo-Ferrer, 2001). For a (multi)set $X = \{\!\{x_1, \ldots, x_p\}\!\}$ with $x_i \in \mathbb{R}^d$ let us denote with $X_{(i)} = \{\!\{x_1 \cdot e_i, \ldots, x_p \cdot e_i\}\!\}$ the multi-set of values projected onto the $i$-th standard basis vector.

**Theorem 12** (Generalizing the result from Oganian & Domingo-Ferrer (2001)). *Microaggregation for cost functions $C^{(d)}(X) := \sum_{i=1}^d C(X_{(i)})$ is NP-hard in dimensions $d \geq 2$ and $k = 3$ if splitting is beneficial for cost function $C$, and the ratios of minimal structure cost are $\frac{l_4(C)}{l_3(C)} > \frac{4}{3}$, and $\frac{l_5(C)}{l_3(C)} > \frac{5}{3}$.*

Computing the $l_i, i \in \{3, 4, 5\}$ values for the cost functions in corollary 7, we get the following:

**Corollary 13.** *Microaggregation is NP-hard in dimensions $\geq 2$ and $k = 3$ for the cost functions SSE, SAE, and Round up/Round down cost.*

The original result by Oganian & Domingo-Ferrer (2001) dealt with SSE only but the proof allows easy adaption for SAE, and round up/round down cost. Unfortunately, theorem 12 is not applicable to the maximum distance cost as both $l_i$ ratios are too small.

## B.2    Total monotonicity and concave cost functions

### B.2.1    Monge matrices and their transpose are totally monotone

Before we start with the proof for concavity we recap the relationship of concave cost functions and totally monotone matrices. For a concave cost function we start by rearranging the quandrangle inequality eq. (1) as

$$C(a, c) - C(a, d) \leq C(b, c) - C(b, d) \text{ for } 0 \leq a < b < c < d \leq n$$

We then notice that now $C(b, c) - C(b, d) < 0 \Rightarrow C(a, c) - C(a, d) < 0$. If we now consider the matrix $\mathbf{C}$ with entries $\mathbf{C}_{ij} = C(i, j)$, all 2 by 2 submatrix of $\mathbf{C}$ have the form

$$\begin{pmatrix} C(a, c) & C(a, d) \\ C(b, c) & C(b, d) \end{pmatrix}$$

for suitable choice of of the integers $a < b, c < d$. To check for monotonicity we see that $C(b, c) < C(b, d)$ which is equivalent to $C(b, c) - C(b, d) < 0$ which by our previous equality implies $C(a, c) - C(a, d) < 0 \Leftrightarrow C(a, c) < C(a, d)$. That means that any 2 by 2 submatrix is totally monotone. Thus the entire matrix is totally monotone.

For the transpose $\mathbf{C}^T$ we find that 2 by 2 submatrices are of the form

$$\begin{pmatrix} C(a, c) & C(b, c) \\ C(a, d) & C(b, d) \end{pmatrix}.$$

To check for total monotonicity we need to show that $C(a, d) < C(b, d) \Leftrightarrow C(a, d) - C(b, d) < 0$ implies $C(a, c) < C(b, c) \Leftrightarrow C(a, c) - C(b, c) < 0$. By rearranging the quadrangle inequality we find

$$C(a, c) - C(b, c) \leq C(a, d) - C(b, d) \text{ for } 0 \leq a < b < c < d \leq n.$$

Now $C(a, d) - C(b, d) \leq 0 \Rightarrow C(a, c) - C(b, c)$ which is exactly what we need such that any 2 by 2 submatrix of $\mathbf{C}^T$ is totally monotone. Thus the entire matrix $\mathbf{C}^T$ is totally monotone.

### B.2.2   Proof that $\tilde{\mathbf{M}}^T$ is totally monotone

We prove total monotonicity of $\tilde{\mathbf{M}}^T$ by showing that all 2 by 2 submatrices fullfill the necessary equations. We never explicitly consider a submatrix but we choose two entries in the same row but different columns (col1 < col2). We then compare the values of $\tilde{\mathbf{M}}^T$ at those locations (val1:= $\tilde{\mathbf{M}}^T_{\text{row,col1}}$, val2:= $\tilde{\mathbf{M}}^T_{\text{row,col2}}$). Depending on the size of the values, this imposes requirements on either the values below (in case of val1 > val2) or the entries above (in case of val1 $\leq$ val2). If row' is such a row either above or below, we refer to the entries in the same columns as before as val1':= $\tilde{\mathbf{M}}^T_{\text{row',col1}}$, val2':= $\tilde{\mathbf{M}}^T_{\text{row',col2}}$.

$\tilde{\mathbf{M}}^T$ **of the adapted cost in eq. (5)** Instead of referring to explicit values, we refer to the values by their colors. If the cluster corresponding to the entry is too small, the color is red else the color is green (these are the two cases in eq. (5)).

- *Case val1=green, val2 = green:* val1 $\leq$ val2: The pairs above are either green-green, green-red, or red-red. For green-green pairs monotonicity is guaranteed by eq. (1). This implies total monotonicity as the transpose of monge matrices are also monge Burkard et al. (1996), monge matrices are totally monotone Burkard et al. (1996) and if you add constant values to the columns of totally monotone matrices they remain totally monotone Burkard et al. (1996). For green-red, and red-red pairs by design val1' < val2'. Overall values above col1 are less than or equal to those in col2. val1 > val2: The only case if green-green which is correct by monotonitcity.

- *Case val1=green, val2=red:* The only case is val1 < val2, the pairs above are either green-red or red-red in either cases the values above col1 are smaller than those above col2.

- *Case val1=red, val2=red:* The only case here is val1 < val2: all entries above are red-red as well for which val1' < val2'

$\tilde{\mathbf{M}}^T$ **of the adapted cost in eq. (6)** As before, instead of referring to explicit entries, we refer to the entries by their colors as visible in Figure 3, i.e. the first case in eq. (6) is red, the second orange and the third is green.

- *Case val1=green, val2 = green:* val1 $\leq$ val2: The cases above are the same as in the previous consideration.
  val1 > val2: The pairs below are either green-green, orange-green or orange-orange. For green-green pairs monotonicity is guaranteed as in the val1 $\leq$ val2 case. For orange-green and orange-orange pairs, val1' > val2' by design.

- *Case val1=red, val2=red:* the only case here is val1 < val2: all entries above are red-red as well for which val1' < val2'

- *Case val1=orange, val2=green:* The only case is val1 > val2, the possible pairings below are orange-orange or orange-green. We see that the values below col1 are larger than those below col2.

- *Case val1=orange, val2=red:* The only case is val1 < val2, the pairs above are either orange-red or green-red. In both cases the values above column 1 are smaller than those above column 2.

- *Case val1=green, val2=red:* The only case is val1 < val2, the pairs above are either green-red or red-red in either cases the values above col1 are smaller than those above col2.

- *Case val1=orange, val2=orange:* The only case is val1 > val2, the pairs below are orange-orange. Thus the values below column 1 are larger than those in column 2.

### B.3   Computational complexity of computing cost functions

This section shows, that all the cost functions presented in corollary 7 can be computed in $O(1)$ performing at most $O(n)$ time for preprocessing and using at most $O(n)$ space. Throughout this section we assume that the input data is sorted, i.e. $i < j \Rightarrow x_i \leq x_j$. We also assume that $i < j$ such that $X_{i,j} = \{\!\{x_{i+1}, \ldots, x_j\}\!\}$ is well defined.

### B.3.1 Sum of absolute errors

During preprocessing we compute and store the cummulative sum $s(i) = \sum_{l=1}^{i} x_l$. The computation of that cummulative sum is $O(n)$.

For a (multi-)set $A$ we denote with $(A)_1/(A)_2$ the $\left\lfloor \frac{|A|}{2} \right\rfloor$ smallest/largest elements of $A$. Let $l = \left\lfloor \frac{j-i}{2} \right\rfloor$ then we note, that

$$SAE(i,j) = \sum_{x \in (X_{i,j})_2} x - \sum_{x \in (X_{i,j})_1} x$$
$$= s(j) - s(j - l) - (s(i + l) - s(i))$$

which is $O(1)$ if the $s(i)$ are stored in an array.

### B.3.2 Maximum distance cost

For the maximum distance cost $C_\infty$ we have

$$C_\infty(i,j) = \max_{x \in X_{i,j}} |x - \frac{x_{i+1} + x_j}{2}| = \frac{x_j - x_{i+1}}{2}$$

which is $O(1)$ if the $x_i$ are stored in an array.

### B.3.3 Round down/ round up cost

Similar as for SAE for precomputation we compute the cummulative sum $s(i) = \sum_{l=1}^{i} x_l$, $s(0) = 0$ in $O(n)$ and store all of the $s(i)$ values. Then the round down cost is

$$C_\downarrow(i,j) = \sum_{l=i+1}^{j} x_l - x_{i+1} = s(j) - s(i) - (j - i)x_{i+1}.$$

The round up cost is

$$C_\uparrow(i,j) = \sum_{l=i+1}^{j} x_j - x_l = (j - i)x_j - (s(j) - s(i)).$$

Both are $O(1)$ if the $x_i$ and $s(i)$ are stored in an array.

### B.4 Concavity of cost functions

### B.4.1 Maximum distance cost is concave

In 1D the maximum distance cost is defined as $C_\infty(a,b) = \max_{l=a+1}^{b} |x_l - \overset{\infty}{x}_{a..b}|$ where $\overset{\infty}{x}_{a..b} = \frac{x_b + x_{a+1}}{2}$ is chosen such that it minimizes the max expression. Thus in 1D, the maximum distance cost can be written as $C_\infty(a,b) = |x_b - \overset{\infty}{x}_{a..b}| = |x_{a+1} - \overset{\infty}{x}_{a..b}| = \frac{x_b - x_{a+1}}{2}$. We can now show, that the maximum distance cost is concave: $C_\infty(a,c) + C_\infty(b,d) = \frac{x_c - x_{a+1}}{2} + \frac{x_d - x_{b+1}}{2} = \frac{x_d - x_{a+1}}{2} + \frac{x_c - x_{b+1}}{2} = C_\infty(a,d) + C_\infty(b,c)$.

### B.4.2 Round up/round down costs are concave

In 1D the round up cost is $C_\uparrow(a,b) = \sum_{l=a+1}^{b} |x_b - x_l| = (b-a)x_b - \sum_{l=a+1}^{b} x_l$. We can now show, that the round up cost is concave:

$$C_\uparrow(a,c) + C_\uparrow(b,d) \le C_\uparrow(a,d) + C_\uparrow(b,c)$$

$$\Leftrightarrow (c-a)x_c + (d-b)x_d - \sum_{i=a+1}^{c} x_i - \sum_{i=b+1}^{d} x_i$$

$$\le (d-a)x_d + (c-b)x_c - \sum_{i=a+1}^{d} x_i - \sum_{i=b+1}^{c} x_i$$

$$\Leftrightarrow (c-a)x_c + (d-b)x_d \le (d-a)x_d + (c-b)x_c$$

$$\Leftrightarrow -ax_c - bx_d \le -ax_d - bx_c$$

$$\Leftrightarrow (b-a)x_c \le (b-a)x_d$$

$$\Leftrightarrow x_c \le x_d$$

which is true as the points are ordered in increasing order.

We can do a similar argument for the round down cost $C_\downarrow(a,b) = \sum_{i=a+1}^{b} |x_i - x_a| = \sum_{i=a+1}^{b} x_i - (b-a)x_a$.

$$C_\downarrow(a,c) + C_\downarrow(b,d) \le C_\downarrow(a,d) + C_\downarrow(b,c)$$

$$\Leftrightarrow \sum_{i=a+1}^{c} x_i + \sum_{i=b+1}^{d} x_i - (c-a)x_{a+1} - (d-b)x_{b+1}$$

$$\le \sum_{i=a+1}^{d} x_i - \sum_{i=b+1}^{c} x_i - (d-a)x_{a+1} - (c-b)x_{b+1}$$

$$\Leftrightarrow -(c-a)x_{a+1} - (d-b)x_{b+1} \le -(d-a)x_{a+1} - (c-b)x_{b+1}$$

$$\Leftrightarrow -cx_{a+1} + -dx_{b+1} \le -dx_{a+1} - cx_{b+1}$$

$$\Leftrightarrow (d-c)x_{a+1} \le (d-c)x_{b+1}$$

$$\Leftrightarrow x_{a+1} \le x_{b+1}$$

which is true as the points are sorted in increasing order.

### B.4.3 SAE cost is concave

For a (multi-)set $A$ let us denote with $(A)_{1/2}$ the $\left\lfloor \frac{|A|}{2} \right\rfloor$ smallest/largest elements. For brevity let us define $s(A) := \sum_{x \in A} x$. Let $A, B, C$ be the three sets obtained by the cut indicies $a, b, c, d$ ($A = \{\!\{x_{a+1}, \ldots, x_b\}\!\}$, $B = \{\!\{x_{b+1}, \ldots, x_c\}\!\}$, $C = \{\!\{x_{c+1}, \ldots, x_d\}\!\}$). Then quadrangle inequality for SAE reads:

$$C_\downarrow(a,c) + C_\downarrow(b,d) \le C_\downarrow(a,d) + C_\downarrow(b,c)$$

$$\Leftrightarrow -s((A \cup B)_1) + s((A \cup B)_2) - s((B \cup C)_1) + s((B \cup C)_2)$$

$$\le -s((A \cup B \cup C)_1) + s((A \cup B \cup C)_2) - s((B)_1) + s((B)_1)$$

To simplify the above expression we observe the following relations:

$$(A \cup B)_1 \subseteq (A \cup B \cup C)_1 \qquad R_{ABC-AB,1} := (A \cup B \cup C)_1 \setminus (A \cup B)_1$$
$$(B \cup C)_2 \subseteq (A \cup B \cup C)_2 \qquad R_{ABC-BC,2} := (A \cup B \cup C)_2 \setminus (B \cup C)_2$$
$$(B)_1 \subseteq (B \cup C)_1 \qquad R_{BC-B,1} := (B \cup C)_1 \setminus (B)_1$$
$$(B)_2 \subseteq (A \cup B)_2 \qquad R_{AB-B,2} := (A \cup B)_2 \setminus (B)_2$$

Then we have

$$C_\downarrow(a,c) + C_\downarrow(b,d) \le C_\downarrow(a,d) + C_\downarrow(b,c)$$
$$\Leftrightarrow s(R_{ABC-AB,1}) + s(R_{AB-B,2}) \le s(R_{ABC-BC,2}) + s(R_{BC-B,1})$$

which is true as $s(R_{ABC-AB,1}) \le s(R_{BC-B,1})$ and $s(R_{AB-B,2}) \le s(R_{ABC-BC,2})$. One can see that the first expression is true by increasing the size of $A$ gradually. If $|A| = 0$ then certainly the lhs and rhs are the same. If we now add elements to $A$ which are no larger than those in $B$ then the lhs expression will only get smaller. One can employ a similar trick for the second expression by adding elements to $C$ which no smaller than those in $B$, which only increase the rhs expression.

### B.4.4  SSE cost is concave

We already explained how to compute the SSE cost in $O(1)$ using $O(n)$ for preprocessing in section 4.

## B.5  Ordered minimizable

### B.5.1  Proof of Theorem 6

Before we start with the actual proof lets provide some clarifications regarding multi-sets. When calling for the $l < |X|$ smallest elements of a multi-set $X$, we mean the multi-set $S \subseteq X$ of size $l$ which fullfills $\forall_{s \in S} \forall_{x \in (X \setminus S)} s \le x$.

We conduct the proof of theorem 6 in two steps. First we show that q-partition ordered minimizable for all q implies ordered minimizable. We then show that if a cost is 2 partition ordered minimizable it is 2-partition ordered minimizable for all $q$.

**Lemma 14.** *If a cost function is q-partition ordered minimizable for all q then it is ordered minimizable.*

We prove lemma 14 by contradiction. Lets assume that it is not enough to consider only pairwise ordered partitions when finding any partition minimizing a total cost which is q-partition ordered minimizabe for all q. But then there is a partition $P$ (wlog. $|P| = q$) which minimizes the total cost $TC$ subject to the either the UM or k-means constraint. But then as the total cost $TC$ is q-partition ordered minimizable we are guaranteed a pairwise ordered partition $P'$ with has the same multiset of sizes as $P$ but no larger total cost. Thus $P'$ also respects the UM or k-means constraint. Thus it would be enough to have considered only pairwise ordered partitions when minimizing the total cost on the relevant partitions, which concludes our contradiction.

*Proof.* Now we show that if a cost is 2-partition ordered minimizable, then it is q-partition ordered minimizable for all $q$ by induction on $q$.

The base case is the assumption. Assume the TC is q-ordered minimizable. Let $P = \{\!\{X_0, \ldots, X_{q+1}\}\!\}$ be a (q+1)-partition. We can construct a pairwise ordered partition $P'$ which has no worse TC than $P$: Let $A_1 = X_1$. From 2-partition ordered minimizable we know that we can obtain sets $B_{i+1}, C_{i+1} = \text{pairwise\_order}(A_i, X_{i+1})$ (see corollary 7). We then define $A_{i+1} := \min(B_{i+1}, C_{i+1})$ and $\tilde{X}_i := \max(B_{i+1}, C_{i+1})$. Note that $A_{i+1}$ contains the smallest elements in $\bigcup_{j=1}^{i+1} X_j$. We can then define the partition

$$\tilde{P}_i := \{\!\{\tilde{X}_1, \ldots, \tilde{X}_{i-1}, A_i, X_{i+1}, \ldots, X_{q+1}\}\!\}.$$

We see that $\tilde{P}_1 = P$ and $TC(\tilde{P}_{i+1}) \leq TC(\tilde{P}_i)$. The latter is consequence from

$$TC(\tilde{P}_{i+1}) = \sum_{j=1}^{i} C(\tilde{X}_j) + C(A_{i+1}) + \sum_{j=i+2}^{q+1} C(X_j)$$

$$\leq TC(\tilde{P}_i) = \sum_{j=1}^{i-1} C(\tilde{X}_j) + C(A_i) + \sum_{j=i+1}^{q+1} C(X_j)$$

which is equivalent to

$$C(\tilde{X}_i) + C(A_{i+1}) \leq C(A_i) + C(X_{i+1}).$$

Here we note, that either $|A_{i+1}| = |X_{i+1}|$ and $|\tilde{X}_i| = |A_i|$ or $|A_{i+1}| = |A_i|$ and $|\tilde{X}_i| = |X_{i+1}|$. In either case the equations can be written as

$$TC(\{L, R\}) \leq TC(\{Y, Z\}) \tag{9}$$

with $|L| = |Y|$, $|R| = |Z|$ and $L, R$ being pairwise ordered but this is exactly being 2-partition ordered minimizable which we have by assumption. Overall we have now found a partition $\tilde{P}_{q+1} = \{\{\tilde{X}_1, \ldots, \tilde{X}_q, A_{q+1}\}\}$ with TC no worse than the TC of $P$. Now we consider the minimization problem of TC on $\Omega \setminus A_{q+1}$ subject to partitions of size $q$. By the induction hypothesis we are guaranteed a pairwise ordered partition $\{\{\hat{X}_1, \ldots, \hat{X}_q\}\}$ that minimizes TC on $\Omega \setminus A_{q+1}$. Now we have found a (q+1) ordered partition $P' = \{\{\hat{X}_1, \ldots, \hat{X}_q, A_{q+1}\}\}$ which has the same size multiset and no worse TC than $P$ but is totally ordered (remember the $\hat{X}_i$ are pairwise ordered and all elements in $A_{q+1}$ are no larger than any elements in any of the $\hat{X}_i$). This concludes the induction. □

### B.5.2 Ordered minimizable and quadrangle inequality

In this subsection we provide evidence that quadrangle inequality and ordered minimizable are indeed not the same concepts by providing a simple counter example.

**Lemma 15.** *Quadrangle inequality does not imply ordered minimizable.*

*Proof.* Let our universe $\Omega \subseteq \mathbb{R}^{\geq}$ be a finite set and

$$C_{\min}(X) := \min(X).$$

Then by the following $C_{\min}$ fullfills the quadrangle inequality ($a < b < c < d$)

$$C_{\min}(a, c) + C_{\min}(b, d) \leq C_{\min}(a, d) + C_{\min}(b, c)$$

$$\Leftrightarrow x_{a+1} + x_{b+1} \leq x_{a+1} + x_{b+1}$$

The last line is certainly true.

We will now show, that $C_{\min}$ is not ordered minimizable. Let $|\Omega| \geq 4$ and let $\omega_1 < \omega_2 < \omega_3$ be the smallest, second smallest, and third smallest element of the universe $\Omega$. Let the multi-set $R := \Omega \setminus \{\omega_1, \omega_2, \omega_3\}$. Then for $A = \{\omega_1, \omega_3\}$ and $B = \{\omega_2\} \cup R$ be two sets then $TC(\{A, B\}) = \omega_1 + \omega_2$ but in every ordered partition with $\min(|L|, |R|) \geq 2$ we have that $\omega_1$ and $\omega_2$ are in the same set. But this means for any pairwise ordered sets $L$ and $R$ with $\min(|L|, |R|) \geq 2$ we have $TC(\{L, R\}) \geq \omega_1 + \omega_3 > TC(\{A, B\})$. Thus $C_{\min}$ is not ordered minimizable. □

### B.5.3 SSE is 2-partition ordered minimizable

Looking at the variance $\mathrm{var}(X) = \frac{1}{|X|} \sum_{x \in X} (x - \bar{X})^2 = \frac{1}{|X|} \left( \sum_{x \in X} x^2 \right) - \bar{X}^2$ where the latter is a common simplification. We now note, that $\mathrm{SSE}(X) = n \, \mathrm{var}(X) = \left( \sum_i x_i^2 \right) - n\bar{X}^2$ thus when comparing two partitions

$P_1$ and $P_2$ of the same original set we see, that

$$\text{TSSE}(P_1) \leq \text{TSSE}(P_2)$$
$$\Leftrightarrow -\sum_{X \in P_1} n_X \bar{X}^2 \leq -\sum_{X \in P_2} n_X \bar{X}^2$$

For the case of two partitions $P_1 = \{\!\{A + \tilde{A}, B + \tilde{B}\}\!\}$ and $P_2 = \{\!\{A + \tilde{B}, B + \tilde{A}\}\!\}$ with $|\tilde{A}| = |\tilde{B}|$ but $\tilde{A} \neq \tilde{B}$ (think of exchanging elements $\tilde{A}$ from $A + \tilde{A}$ with the $\tilde{B}$ elements in $B + \tilde{B}$). This means

$$\text{TSSE}(P_1) \leq \text{TSSE}(P_2)$$
$$\Leftrightarrow -n_{A+\tilde{A}}\mu^2_{A+\tilde{A}} - n_{B+\tilde{B}}\mu^2_{B+\tilde{B}} \leq -n_{A+\tilde{B}}\mu^2_{A+\tilde{B}} - n_{B+\tilde{A}}\mu^2_{B+\tilde{A}}$$
$$\Leftrightarrow n_{A+\tilde{A}}\mu^2_{A+\tilde{A}} - n_{A+\tilde{B}}\mu^2_{A+\tilde{B}} \geq n_{B+\tilde{A}}\mu^2_{B+\tilde{A}} - n_{B+\tilde{B}}\mu^2_{B+\tilde{B}}$$

Where $n_S$ denotes the size of set $S$ and $\mu_S$ denotes the average of the set $\mu$. To treat the lhs and rhs simultaneously, lets consider multisets $X, Y, Z$ with $|Y| = |Z|$:

$$n_{X+\tilde{Y}}\mu^2_{X+\tilde{Y}} - n_{X\tilde{Z}}\mu^2_{X+\tilde{Z}}$$
$$= \frac{1}{n_{X+\tilde{Y}}}\left(s_X^2 + s_Y^2 + 2s_X s_Y - s_X^2 - s_Z^2 - 2s_X s_Z\right)$$
$$= \frac{1}{n_{X+\tilde{Y}}}\left(s_Y^2 + 2s_X s_Y - s_Z^2 - 2s_X s_Z\right)$$
$$= \frac{1}{n_{X+\tilde{Y}}}\left(s_Y - s_Z\right)\left(s_Y + s_Z + 2s_X\right)$$
$$= \left(s_Y - s_Z\right)\left(\mu_{X+Y} + \mu_{X+Z}\right)$$

Bringing this together with the previous equations yields

$$\text{TSSE}(P_1) \leq \text{TSSE}(P_2)$$
$$\Leftrightarrow (s_{\tilde{A}} - s_{\tilde{B}})(\mu_{A+\tilde{A}} + \mu_{A+\tilde{B}}) \geq (s_{\tilde{A}} - s_{\tilde{B}})(\mu_{B+\tilde{A}} + \mu_{B+\tilde{B}})$$
$$\Leftrightarrow \mu_{A+\tilde{A}} + \mu_{A+\tilde{B}} \leq \mu_{B+\tilde{A}} + \mu_{B+\tilde{B}}$$
$$\Leftrightarrow \mu_{A+\tilde{B}} - \mu_{B+\tilde{A}} \leq \mu_{B+\tilde{B}} - \mu_{A+\tilde{A}}$$

Where the the second last line is true if $s_{\tilde{A}} < s_{\tilde{B}}$. If we can maximize the right hand side of the last expression which is only a function of the partition $P_1$, this is the same as minimizing the TSSE of $P_1$ with respect to $P_2$. The maximum of the rhs is achieved by choosing that $B$ and $\tilde{B}$ contain the largest elements while $A$ and $\tilde{A}$ contain the smallest elements. This naturally fullfills $s_{\tilde{A}} < s_{\tilde{B}}$ as $|\tilde{A}| = |\tilde{B}|$ (remember $\tilde{A} \neq \tilde{B}$). Thus we conclude that for all 2-partitions $P_2$ we can find pairwise orderd $P_1$ which has no worse TSSE, i.e. TSSE is 2-partition ordered minimizable.

### B.5.4 Infinity norm is 2-partition ordered minimizable

We denote the smallest/largest element in $\Omega$ with $l/r$. Lets assume we are given a partition $P$ of the elements in $\Omega$ into $A$ and $B$. Wlog lets assume $l \in A$. We choose two sets $L$ and $R$ such that $L$ contains the $|A|$ smallest elements and $R$ the remaining elements, then

$$C_\infty(L) + C_\infty(R) \leq C_\infty(A) + C_\infty(B)$$
$$\Leftrightarrow \frac{\max(L) - l}{2} + \frac{r - \min(R)}{2} \leq \frac{\max(A) - l}{2} + \frac{\max(B) - \min(B)}{2}$$
$$\Leftrightarrow \max(L) - \min(R) \leq \begin{cases} \max(A) - \min(B) & \text{if } \max(B) = r \\ \max(B) - \min(B) & \text{if } \max(A) = r \end{cases}$$

In the first case, we have that $\max(L) \leq \max(A)$ as $L$ contains the smallest elements and $|A| = |L|$. Further, $\min(B) \leq \min(R)$ as $R$ contains the largest elements and $|B| = |R|$. In the second case we observe that the left hand side is always non positive while the right hand side is always non negative.

### B.5.5 SAE is 2-partition ordered minimizable

For a multiset $A$ we denote with $a_1/a_2$ the multisets of size $\left\lfloor \frac{|A|}{2} \right\rfloor$ containing the smallest/largest elements of $A$. We define $b_i, l_i, r_i$ similar for multisets $B, L, R$. We then note, that $SAE(A) = \sum_{x \in a_1} -x + \sum_{x \in a_2} x$ irrespective of whether $|A|$ is even or odd. Lets consider a partition of the elements in $\Omega$ into two sets $A, B$. Let us further the multisets $c_{1/2} := l_{1/2} \cup r_{1/2}$ and $d_{1/2} := a_{1/2} \cup b_{1/2}$. In a slight abuse of notation let us denote the index (i.e. position in an order of $\Omega$) of the i-th smallest element in $c_{1/2}$ as $c_{1/2}(i)$. Then $x_{c_{1/2}(1)}$ is the smallest, $x_{c_{1/2}(2)}$ is the second smallest element of $c_{1/2}$ and so on. We similarly do that with the sets $d_{1/2}$ (i.e. $x_{d_{1/2}(i)}$ is the i-th smallest element of $d_{1/2}$). We define $L/R$ as the sets containing the smallest/largest elements of $\Omega$. The size of $L$ is $\min(|A|,|B|)$ if $d_2(1) \leq \max(|a_1|,|b_1|) + 1$ and $\max(|A|,|B|)$ otherwise. The size of $R$ is $|\Omega| - |A|$.

Lets us first consider the first $|l_1|$ of the sets $c_{1/2}$ and $d_{1/2}$. For $i \leq |l_1|$ certainly $c_1(i) = d_1(i)$ (these are the $|l_1|$ smallest elements of $\Omega$). Further, $c_2(i) \leq d_2(i)$ as $c_2(1) \leq d_2(1)$ and the $c_2$ continue consecutively for the next $|l_1| - 1$ elements. In the case $|L| = \min(|A|,|B|)$ the relation $c_2(1) \leq d_2(1)$ is pretty obvious as in this case $c_2(1)$ is minimal among all partitions with the same size. In the other case, we use that now $d_2(1) > \max(|a_1|,|b_1|) + 1 \geq c_2(1)$.

For the remaining $|r_1|$ elements in $c_{1/2}$ or $d_{1/2}$, let $i \leq |r_1|$ and we can do a similar argument but from the other direction. For ease of notation we set $n_{\frac{1}{2}} := |l_1| + |r_1|$. We observe that $c_2(n_{\frac{1}{2}} - i) = d_2(n_{\frac{1}{2}} - i)$ as these are the $|r_1|$ largest elements. Also $c_1(n_{\frac{1}{2}} - i) \geq d_1(n_{\frac{1}{2}} - i)$ as $c_1(n_{\frac{1}{2}}) \geq d_1(n_{\frac{1}{2}})$ and the next $|r_1| - 1$ smaller elements are consecutively. In the case $|L| = \max(|A|,|B|)$ the relation $c_1(n_{\frac{1}{2}}) \geq d_1(n_{\frac{1}{2}})$ is pretty obvious as in this case $c_1(n_{\frac{1}{2}})$ is maximal among all partitions with the same size. For the other case let $|A| \leq |B|$. We use that now $d_2(1) \leq \max(|a_1|,|b_1|) + 1$ which implies that $\max(a_1) \leq d_2(1) - 1 \leq \max(|a_1|,|b_1|)$ or in other words all indices of elements in $d_1$ that are larger than $\max(|a_1|,|b_1|)$ are from $b_1$. Because there are at least $|b_1|$ elements larger than those in $b_1$, we can conclude that $d_1(n_{\frac{1}{2}}) \leq n - |b_1| \leq c_1(n_{\frac{1}{2}})$.

Turning back to the SAE we find that

$$SAE(L) + SAE(R) \leq SAE(A) + SAE(B)$$
$$\Leftrightarrow \sum_{i=1}^{|a_1|+|b_1|} x_{c_2(i)} - x_{c_1(i)} \leq \sum_{i=1}^{|a_1|+|b_1|} x_{d_2(i)} - x_{d_1(i)}$$

Looking at individual terms we find that $x_{c_2(i)} - x_{c_1(i)} \leq x_{d_2(i)} - x_{d_1(i)}$ because $c_2(i) \leq d_2(i)$ and $c_1(i) \geq d_1(i)$. If the individual lhs terms are no larger than the rhs term, then certainly the lhs sum is no larger than the rhs sum.

### B.5.6 Round up/down costs are 2-partition ordered minimizable

Let $\{A, B\}$ be a 2-partition of $\Omega$. Wlog $\max(A) \leq \max(B)$ then certainly $\max(B) = \max(\Omega)$. We now choose $L, R$ to be the smallest/largest elements of $\Omega$ with sizes $|A|$ and $|B|$ respectively. Then the round-up cost is

$$C_\uparrow(L) + C_\uparrow(R) \leq C_\uparrow(A) + C_\uparrow(B)$$
$$\Leftrightarrow \sum_{x \in L} \max(L) - x + \sum_{x \in R} \max(R) - x \leq \sum_{x \in A} \max(A) - x + \sum_{x \in B} \max(B) - x$$
$$\Leftrightarrow |A| \max(L) \leq |A| \max(A)$$

Where a lot of terms cancel as each $x$ appears exactly once, and $\max B = \max R$. Certainly $\max(L) \leq \max(A)$ as both have the same number of elements and $\max L$ is minimal over all sets of that size.

We can do a similar argument for the round down cost. Let $\{A, B\}$ be a 2-partition of $\Omega$. Wlog $\min(A) \leq \min(B)$ then certainly $\min(A) = \min(\Omega)$. We now choose $L, R$ to be the smallest/largest elements of $\Omega$ with

sizes $|A|$ and $|B|$ respectively. Then the round-down cost is

$$C_\downarrow(L) + C_\downarrow(R) \leq C_\downarrow(A) + C_\downarrow(B)$$
$$\Leftrightarrow \sum_{x \in L} x - \min(L) + \sum_{x \in R} x - \min(R) \leq \sum_{x \in A} \min(A) - x + \sum_{x \in B} \min(B) - x$$
$$\Leftrightarrow -|B| \min(R) \leq -|B| \min(B)$$
$$\Leftrightarrow \min(R) \geq \min(B)$$

Which is true as $\min(R)$ is maximal among all sets of its size.

### B.6  Negative results on ordered minimizable

The mean absolute error $(MAE(X) = \frac{1}{|X|} \sum_{x \in X} |x - \mathrm{Median}(X)|)$ is *not ordered minimizable*. One counter example is: For $\Omega = \{\{-1, 0, 0, 0, 0, 1\}\}$, the partition $\{\{\{-1, 1, 0, 0\}\}, \{\{0, 0\}\}\}$ has cost $\frac{1}{2}$ but the two pairwise ordered partitions $\{\{\{-1, 0, 0, 0\}\}, \{0, 1\}\}$ and $\{\{\{-1, 0\}\}, \{\{0, 0, 0, 1\}\}\}$ both have costs $\frac{3}{4}$.

The $\ell_2$ cost $(C_{\ell_2}(X) = \sqrt{SSE(X)})$ is *not ordered minimizable*. One counter example is: For $\Omega = \{\{-1, 0, 0, 0, 1\}\}$, the partition $\{\{\{-1, 1, 0\}\}, \{\{0, 0\}\}\}$ has cost $\sqrt{2} \approx 1.41$ but the two pairwise ordered partitions $\{\{\{-1, 0, 0\}\}, \{0, 1\}\}$ and $\{\{\{-1, 0\}\}, \{0, 0, 1\}\}$ both have costs $\sqrt{\frac{1}{2}} + \sqrt{\frac{2}{3}} \approx 1.52$.

The mean round down error $(d_{\downarrow}(X) = \frac{1}{|X|} \sum_{x \in X} x - \min(X))$ is *not ordered minimizable*. One counter example is: For $\Omega = \{\{0, 0, 0, 1, 1, 2\}\}$, the partition $\{\{\{0, 0, 0, 2\}\}, \{\{1, 1\}\}\}$ has cost $\frac{1}{2}$ but the two pairwise ordered partitions $\{\{\{0, 0, 0, 1\}\}, \{1, 2\}\}$ and $\{\{0, 0\}\}, \{\{0, 1, 1, 2\}\}\}$ have costs $\frac{3}{4}$ and $1$ respectively.

One can construct a similar counter example for mean round up cost. The mean round up error $(d_{\uparrow}(X) = \frac{1}{|X|} \sum_{x \in X} \max(X) - x)$ is *not ordered minimizable*. One counter example is: For $\Omega = \{0, 1, 1, 2, 2, 2\}$, the partition $\{\{\{0, 2, 2, 2\}\}, \{\{1, 1\}\}\}$ has cost $\frac{1}{2}$ but the two pairwise ordered partitions $\{\{\{0, 1, 1, 2\}\}, \{\{2, 2\}\}\}$ and $\{0, 1\}, \{\{1, 2, 2, 2\}\}\}$ have costs $1$ and $\frac{3}{4}$ respectively.

### B.7  Ordered minimizable but splitting is not beneficial

We were wondering whether we could provide a cost function which is ordered minimizable but does not have the property, that splitting is beneficial. Let us denote with $X_{>M(X)}$ the set of elements in the set $X$ larger than median of $X$. Let our universe consist only of non negative reals i.e. $\Omega \subseteq \mathbb{R}^{\geq}$. Then we can define the a cost function

$$C^*(X) = \frac{2}{|X|^\alpha} \sum_{x \in X_{>M(X)}} x.$$

If you consider your universe to be a set of item prices, the cost function describes a scenario where you pay a discounted price on only the most expensive half of items. The discount parameter $0 \leq \alpha \leq 1$ controls the amount of discount provided. Values of $\alpha$ close to 1 indicate that you get a significant discount if you buy in bulk, while low values of $\alpha$ indicate very low discount when buying bulk.

The cost function $C^*(X)$ is ordered minimizable but it does not have the splitting is beneficial property for $\alpha > 0$. For $\alpha = 1$ an optimal UM clustering is just a single cluster containing all points independent of the actual universe $\Omega$.

