# OpenReview forum: "Faster optimal univariate microaggregation"
_TMLR — Accepted by TMLR_

### Review · Reviewer_XfgC · 2024-07-18

**Summary Of Contributions:**

This paper provides an optimal univariate microaggregation on sorted data in O(n) time and space, which is better than prior work regarding time complexity. Experiments show performance improvement.

**Audience:**

Yes

**Claims And Evidence:**

Yes

**Requested Changes:**

Some clarifications needed:

This paper uses synthetic data and claims it is a real-world setting. How do you define the real-world setting for your experiment?

Can you explain your experiment setting for Figures 4 and 5? Did you get the plot by running the experiment once or multiple times? Are you using the top results or the average result?

**Strengths And Weaknesses:**

Strengths:
This paper proposed a novel optimal univariate microaggregation, which offers better time complexity.
The equation and analysis are easy to follow.


Weaknesses:
The data used for the experiment is not clear.  The claim for Real-world improvement does not make sense because this paper uses synthetic data. Where did you get one million reals?

---

> ### Author Response · Authors · 2024-08-02
>
> We wish to thank the reviewer for his comments regarding the clarity of our work.
>
> As the reviewer has pointed out correctly, we use synthetic data in our experiments. In our abstract we write that “the presented algorithms lead to real world performance improvements”. What we meant to say is that the presented algorithms lead to performance improvements on current (commodity) hardware, and is not just a purely theoretical consideration. We will adjust the wording accordingly.
>
> For our experimental setting we average over 10 runs. In the presented plots, we show the mean and the standard deviation of the obtained sample. Thanks for pointing out this omission, we will add this missing information in appropriate places.

---

### Review · Reviewer_yZTt · 2024-08-02

**Summary Of Contributions:**

The paper considers the problem of microaggregation whereby data points are grouped into clusters of at least k points in order to guarantee k-anonymity---a type of privacy guarantee. In particular, the paper improves the time complexity for performing univariate microaggregation from $O(kn)$ to $O(n)$ on sorted data. The approach consists of casting the problem as an instance of the so-called concave last weight subsequence problem and works for several cost functions used for univariate microaggregation. Moreover, the claimed improvement in the asymptotic complexity is also reflected in some experiments.

**Audience:**

Yes

**Broader Impact Concerns:**

No concerns regarding ethical implications.

**Claims And Evidence:**

Yes

**Requested Changes:**

If possible, the authors should expand on the motivation and the significance of the results in the context of my comments above. It would also benefit the paper if further experiments with more realistic data were included.

Some minor comments:

- On 3 occasions you write "we proof" which is incorrect.
- Page 20: there are issues with space before citations.
- There are some missing punctuation marks in the equations of the appendix.

**Strengths And Weaknesses:**

Strengths. The main problem of interest, namely univariate microaggregation, is well-motivated and well-studied in prior literature. Moreover, in the regime where $k$ is large, the obtained improvement in the time complexity can be significant, which is also reflected in the experiments. From a technical standpoint, the approach is non-trivial and combines several interesting ideas starting from the connection with the least weight subsequence problem. Although many of the underlying ideas have been used before, I believe that the overall approach is technically novel and interesting. The main body of the paper does a nice job at explaining adequately the key ideas in the proofs. The writing overall is reasonable and the paper is well-organized. To my knowledge, all relevant papers have been discussed.

Weaknesses. I believe that some crucial aspects of the paper's focus require further justification. The improvement in performance, both in theory and in the experimental results, depends crucially on how large parameter $k$ is. Is it not clear to me how large that parameter is in practical applications; perhaps taking $k$ to be reasonably small suffices for most purposes, in which case the theoretical bounds become rather vacuous (in light of prior work). Perhaps the authors can comment on how large $k$ should be thought of compared to $n$. Taking a step back, it is not even entirely clear how significant the refinement in the time complexity is for a problem such as microaggression; it is an offline problem in which reducing the running time by a small factor does not seem to be particularly consequential for most purposes; again, this depends on the specific applications the paper is targeting, which I feel is not discussed enough. Remaining on the motivation of the paper, it is also a significant caveat that the paper focuses only on the univariate case, which significantly limits the scope of the paper, although this has been acknowledged by the authors. Moreover, the experimental results are not entirely convincing. In particular, experimenting only on data generated under the specific process described in the paper can lead to misleading implications. Perhaps the authors can experiment with some real available data instead of creating such a synthetic dataset.

Besides the issues above, it is not clear whether a machine learning venue is suited for this paper. I feel that both the techniques and the results will be more relevant to a different venue.

---

> ### Author Response · Authors · 2024-08-27
>
> We thank the reviewer for his general comments and his/her constructive feedback.
>
> >  Perhaps the authors can comment on how large $k$ should be thought of compared to $n$
>
> In practice when Microaggregation is applied to census data, the parameter $k$ is usually small less than 10 for datasets with $n\approx 10k$ . In principle, from a privacy standpoint it would be most desirable to choose $k$ as large as possible while maintaining a targeted utility. We would assume that in most situations $k$ may grow slowly with $n$. Lastly, the performance improvements are present even for low values of $k$ through the simple+ algorithm although the main improvements are in the large $k$ regime, as noted correctly by the reviewer.
>
> > microaggression is an offline problem
>
> The reviewer is also correct that the microaggregation problem is usually applied in offline scenarios. We view the presented work more as a theoretical investigation into the microaggregation problem. With some of the presented experiments our main goal was to emphasize that the presented considerations lead to performance improvements on real hardware.
> In practice Multivariate Microaggregation problems are more relevant than Univariate problems, and projection methods, i.e. projecting the multivariate problem into 1D, solving the 1D problem and projecting back to higher D, are a plausible heuristic approach to solve them. Depending on the number of dimensions, solving the 1D problem may take up a significant portion of the runtime. There might thus still be a practical use for presented approaches.
>
> > it is also a significant caveat that the paper focuses only on the univariate case
>
> > It would also benefit the paper if further experiments with more realistic data were included.
>
> We have expanded the experiment section by adding an experiment for multivariate microaggregation applied to two real world dataset. We thereby compare two projection based approaches with one k-means based approaches. We find that for one dataset, the multivariate loss of the projection based approaches is smaller while on the other dataset, the k-means based approach is better. These experiments shed some light on potential applications of present algorithms beyond the univariate case.
>
> > it is not clear whether a machine learning venue is suited for this paper.
>
> Regarding the relevance of the present work to this journal we would consider Microaggregation a clustering task just like k-means but with a different constraint and is thus well suited for a machine learning venue.
>
> We have also fixed the mentioned spelling mistakes and would like to thank the reviewer for spotting them.

---

### Review · Reviewer_JNUo · 2024-08-13

**Summary Of Contributions:**

This paper proposes algorithmic solutions to the univariate microaggregation problem. The authors first reformulated the problem to an instance of the least weight subsequence problem which requires $O(n^2)$ time to solve. The authors then discussed certain properties associated with the cost function which can lower the timing to $O(kn)$ time, and with more properties, one can achieve $O(n)$. Theoretical guarantees are given, along with some experiments highlighting the speed associated with the provided algorithms.

**Audience:**

Yes

**Broader Impact Concerns:**

I have no concerns about the ethical implications of this work, due to it relying on previous works that already addressed the ethical implications of using the microaggregation problem. I would suggest the authors add a small paragraph discussing this at the end of the paper (discuss further the ability of the microaggregation to ensure some privacy measures).

**Claims And Evidence:**

Yes

**Requested Changes:**

* Writing related issues:
  * Change "proof" to "prove" at the third line below equation 6.
  * Add citation to the second line below equation 1.
  * You meant to say lines 15-17 instead of Line 16 at the caption of Figure 3?
  * Right the start of section B.2.2, you meant to say $\tilde{\mathbf{M}}_{row,col2}$ for $val2$?

* Experiment-wise:
   * I understand that it is problematic to apply your approach to the multivariate microaggregation problem, however, I still suggest the authors to run their approach on at least two real-world datasets.

* Questions:
   * In the setting of the problem, to ensure some sort of privacy, the microaggregation problem is given to address this problem. Why did the authors not consider using matrix sketches? for example for the $k$-means problem which in turn gives a smaller set of points that might not even be a subset of the data such that solving the optimal $k$-means on that data would give an approximate solution to the optimal solution obtained when running the same algorithm on the entire data?
   * Is there no way to apply your algorithms to the multivariate microaggregation problem using the projection technique? I understand that this would be considered to be a heuristic more than a provable approach. In case one can, how would you measure the credibility of your approach? I.e., what would be the metric for the effectiveness of your approach aside from running time?

**Strengths And Weaknesses:**

In what follows, the strengths of the paper are listed:

* The univariate microaggregation problem formulation as an instance of the least weight subsequence problem is interesting.
* Inspecting the property of "splitting is beneficial" is interesting, and showing various functions that adhere to such property is indeed eye-opening.
* Utilizing the concavity of certain loss functions led to faster time is also a novel approach.

While the ideas in the paper are combined meticulously, there are some weaknesses with the approaches:
* Some of the writing needs to be addressed (see next section)
* The algorithms seem to work explicitly for the univariate microaggregation problem, while not working directly with the multivariate microaggregation problem due to the sorting-related phase, as well as, losing the concavity property which allows for $O(n)$ time. The authors have mentioned this issue.

---

> ### Author Response · Authors · 2024-08-27
>
> We thank the reviewer for his comments and for spotting the small inaccuracies.
>
>
> > I understand that it is problematic to apply your approach to the multivariate microaggregation problem, however, I still suggest the authors to run their approach on at least two real-world datasets.
>
> In light of the reviewers questions, we did expand the experiment section to include applications of the derived univariate algorithms to the multivariate case through projection (see below).
>
> > Why did the authors not consider using matrix sketches?
>
> In 1D the only matrix sketch we could think of are sub-sampling approaches, but it is not obvious how the minimum group size constraint is respected on the subsampled dataset.
>
> For the multivariate case, we have extended the experiments to included a simple PCA based matrix sketching approach as a possible algorithm to solve the multivariate task. This will certainly only provide an approximate solution to the multivariate case, and PCA serves as a most simple form of a matrix sketching approach.
> The included matrix sketching approach works by using PCA to project the dataset into 1D, solve the univariate problem to obtain a clustering, and use the optimal 1D clustering as a clustering for the multivariate problem.
>
> > Is there no way to apply your algorithms to the multivariate microaggregation problem using the projection technique?
>
> Regarding the second question posed by the reviewer addressed at the multivariate case, we have conducted some experiment on real world datasets for multivariate microaggregation (MM). We therefore benchmark a PCA projection, a random projection and a k-means+cleanup based clustering approach on two real world dataset (The datasets are called EIA and Tarragona and were used in previous works on MM). All three only provide heuristic solutions to the MM problem and their performance varies wildly across the datasets considered.
>
> > How would you measure the credibility of your approach?
>
> We assess the quality of the multivariate Microaggregation through the multivariate loss. For the first dataset (EIA) we find that the k-means based approach (without projection) greatly outperforms the projection based approaches while for the other dataset (Tarragona) the projection based approaches perform better than the k-means based approach. As we can see from these preliminary experiments that projecting and solving the 1D problem can be a useful primitive for solving the MM problem. However the exact performance will depend on the dataset and setup but is beyond the scope of this work.
>
> >  I would suggest the authors add a small paragraph discussing this at the end of the paper
>
> We will add such a paragraph at the end of the paper.

---

### Decision · Action_Editor_MHSq · 2024-09-20

**Recommendation:** Accept with minor revision

**Comment:**

Requested changes:

My first point is that differential privacy is not mentioned anywhere in the paper. DP is the golden standard of privacy in the ML community, so I think it would be appropriate to have some discussion on how the $k$-anonymity type of protection the microaggregation gives would relate to the privacy protection given by DP.

In the conclusions it is written that: "It is well known, that the privacy concept of k-anonymity may provide insufficient protection if for example the confidential attributes are not sufficiently diverse. Caution should thus be taken if k-anonymity, or microaggregation to achieve k-anonymity, is applied in practice." I think it would make sense to add DP e.g. there.

Please add some discussion on how the microaggregation technique is useful in ML. Here are few example questions that I think could be addressed: How would the privacy guarantees given by microaggregation be useful in ML? What are the pros / cons when compared to ML models trained with DP guarantees?

It was also commented by reviewers JNUo and yZTt that the method seems to be restricted to univariate data. There has been an addition of an experiment where the method is used for projected multivariate data, however it is difficult to judge based on the results how well the method fits to the multivariate case (also because the method works well for one of the datasets, the EIA dataset, but not for the other one, the Tarragona dataset, see Fig. 6). If possible add a comparison to a method designed for multivariate data, such as the one in

Domingo-Ferrer, J., Sebé, F., & Solanas, A. (2008). A polynomial-time approximation to optimal multivariate microaggregation. Computers & Mathematics with Applications, 55(4), 714-732.

Please try to also polish the new material on the multivariate experiment.

**Audience:**

Considering that a large part of TMLR audience is interested in data privacy and privacy-preserving ML, I think this topic would be interesting to TMLR readers. However, similarly to the reviewer yZTt, I also think there is an ML-component could be stronger in the paper. I feel that the additional experiments that use PCA for the multivariate data are in the right direction but more could be done in this respect.

**Claims And Evidence:**

The paper proposes a faster method to carry out microaggregation of univariate data. Microaggregation is a $k$-anonymity type of statistical data disclosure technique that groups $n$ data elements into clusters of size at least $k$. The previous methods for microaggregation of univariate data have the computational complexity $O(kn)$ and the main contribution of this paper is a method which has the complexity $O(n)$.